# BTC-LLM: Efficient Sub-1-Bit LLM Quantization via Learnable Transformation and Binary Codebook

## Abstract

Binary quantization represents the most extreme form of large language model (LLM) compression, reducing weights to $\pm 1$ for maximal memory and computational efficiency. While recent sparsity-aware binarization methods achieve sub-1-bit compression by pruning redundant binary weights, they suffer from three critical challenges: performance deterioration, computational complexity from sparse mask management, and limited hardware compatibility. In this paper, we present BTC-LLM, a novel sub-1-bit LLM quantization framework that leverages weight transformation and binary pattern clustering to overcome these limitations, delivering both superior accuracy and efficiency. Our approach incorporates two key innovations: (1) a Flash and Accurate Binary Codebook that identifies recurring binary vector clusters, compressing them into compact indices with tailored distance metrics and sign-based centroid updates; (2) a Learnable Transformation that optimizes invertible scaling and rotation matrices to align binarized weights with full-precision distributions, enabling incoherence processing to enhance layer-wise representation quality. This eliminates the need for sparse masks, enabling efficient inference on standard hardware. Extensive evaluations across LLaMA-1/2/3, Qwen-2.5/3, and FBI-LLM families demonstrate that BTC-LLM establishes a new state-of-the-art for extreme LLM compression at 1.11∼0.7 bits. Notably, our BTC-LLM delivers strong performance under extreme compression settings, with just a 3.1% accuracy drop on LLaMA-2-13B at 0.8 bits in zero-shot benchmarks while achieving a 1.6× speedup over FP16. Code is in the Appendix.

## 1 Introduction

Recent Large Language Models (LLMs) such as GPT-4o (OpenAI, 2024) and DeepSeek-R1 (Guo et al., 2025) have revolutionized natural language processing (NLP), achieving state-of-the-art performance across diverse tasks (Wei et al., 2022). However, the massive scale of models like DeepSeek-R1 (671B parameters) creates unsustainable memory and storage requirements, preventing practical deployment in constrained environments. Model quantization (Ma et al., 2024b) addresses this by reducing numerical precision (*e.g.*, 4-bit or 8-bit integers), slashing memory usage by 4∼8× with minimal accuracy loss. Recent advances, such as Omniquant (Shao et al., 2023) and DuQuant (Lin et al., 2024a) for post-training quantization, demonstrate that even sub-4-bit methods can maintain > 90% of original model performance.

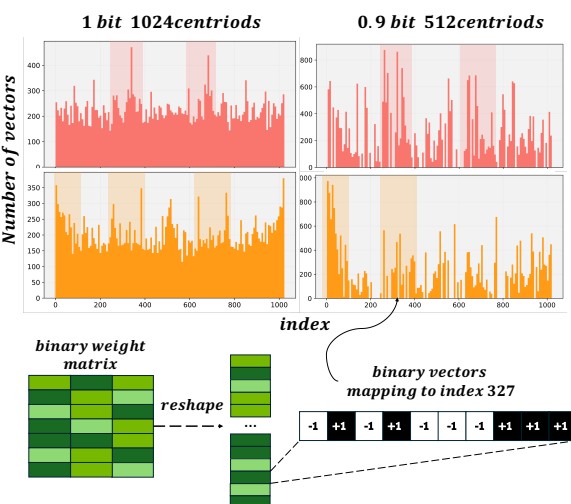

Figure 1: Binary vector distribution (length 10). **Left**: Standard mapping to 1024 indices. **Right**: 512 codebook centroids.

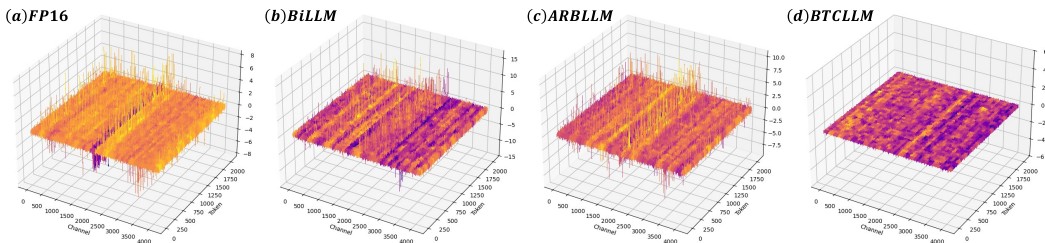

Figure 2: Activation distributions for the `self_attn.k_proj` layer in the LLaMA-2-7B model: (a) Original FP16 (max abs: 8), (b) BiLLM (max abs: 15), (c) ARB-LLM (max abs: 10), and (d) our proposed BTC-LLM (max abs: 0.4).

Binary quantization (Rastegari et al., 2016b) represents the most aggressive quantization approach in the line of model quantization, converting floating-point weights to binary values ($\pm1$) to reduce memory requirements by over $32\times$ (Liu et al., 2018). For instance, BitNet (Wang et al., 2023) pioneered quantization-aware training for 1-bit LLMs, achieving ultra-low memory consumption (0.4GB) and fast inference (29ms). Recent post-training methods like BiLLM (Huang et al., 2024a) and ARB-LLM (Li et al., 2025) employ advanced binarization strategies (*e.g.*, residual approximation, alternating refinement) to enhance 1-bit LLM performance without requiring retraining. STBLLM (Dong et al., 2025) identifies and removes redundant binary parameters to achieve sub-1-bit compression with semi-structured N:M sparsity. However, such sparsity-based binarization faces critical challenges: **(1) Performance Collapse:** STBLLM relies on detecting which elements to prune, yet it suffers from severe accuracy degradation across various LLMs, retaining only 51~65% of full-precision performance on challenging benchmarks (*e.g.*, ARC-c and HellaSwag). **(2) Hardware Incompatibility:** Structured sparsity such as 2:4 is not a free lunch. For instance, each 4-value tuple in a 2:4 pattern admits `C(4,2)=6` possible mask configurations, which requires $\lceil \log_2 6 \rceil = 3$ bits to encode. Consequently, the effective storage cost per weight is $\frac{\text{sign bits+mask bits}}{\text{#weights}} = \frac{2+3}{4} = 1.25$ `bits`. These naturally yield a question:

*(RQ) How can we design a hardware-friendly algorithm to further compress binary weights for sub-1-bit LLMs while maintaining performance?*

To answer this question, we first analyze the weight distribution patterns of binarized LLMs to explore their potential for more compact compression. As shown in Figure 1, we adopt product quantization by splitting the binary weight matrix into sub-vectors, each mapped to an index (e.g., index 327 corresponds to the binary pattern [-1, +1, -1, +1, ...]). Interestingly, these locally continuous blocks exhibit clear clustering patterns, which motivates us to further compress the model by representing redundant $\pm1$ weights with a compact set of centroid vectors.

We further examine the activation distribution of binarized LLMs and empirically observe the presence of prominent outliers. Such large activations amplify the quantization error, since the forward error term can be expressed as $XW - X\widehat{W} = X(W - \widehat{W})$, where outlier entries in $X$ magnify the impact of binarized weight noise. As shown in Figure 2 (b-c), BiLLM shows a wide dynamic range (with absolute values up to 15) with prominent outliers, while ARB-LLM still exhibits noticeable noise and instability. This motivates the need for outlier mitigation, even in binarization methods.

Building on these insights, we propose BTC-LLM, a novel framework that enables extreme compression of LLMs to below 1 bit per parameter. Our approach adopts a two-pronged strategy to tackle key challenges. **First, to exploit redundancy in binary weights, we develop a Flash and Accurate Binary Codebook**, offering a hardware-efficient alternative to sparsity-aware methods that achieves sub-1-bit compression. Our binary-specific codebook compression achieves a superior compression ratio of approximately $16 \cdot v / \lceil \log_2 c \rceil$, where $v$ denotes the vector dimension and $c$ the codebook size. It preserves model performance by retaining the essential distributional characteristics of binary weights. In contrast to sparse quantization, which requires specialized hardware support for efficient N:M sparse computation patterns, our codebook approach enables seamless deployment on standard hardware through simple lookup operations. **Second, to mitigate activation outliers, we introduce a Learnable Transformation** consisting of an invertible parameter $\Lambda$, $D_{\pm}$ and $R$. As shown in Figure 2 (d), this approach effectively suppresses activation outliers, constraining the maximum absolute value to 0.4.

As shown in Figure 3, our comprehensive evaluations of the LLaMA family of models (7B to 65B parameters) demonstrate the superior performance of BTC-LLM in multiple bit width settings.

First, we establish a strong binary baseline that achieves a perplexity of 6.06, surpassing even 2-bit quantization methods. In aggressive quantization regimes (0.9 and 0.8 bits), BTC-LLM exhibits remarkable robustness, maintaining performance nearly identical to its 1.11-bit configuration. Even at 0.7 bits, it achieves a reasonable perplexity of 11.02, while attaining a $22\times$ reduction in memory usage. In zero-shot benchmarks, BTC-LLM consistently outperforms STBLLM by significant margins across model sizes, with gains of +4.7% on LLaMA-1-13B and +5.0% on LLaMA-2-13B at 0.8 bits, demonstrating exceptional robustness under sub-bit quantization.

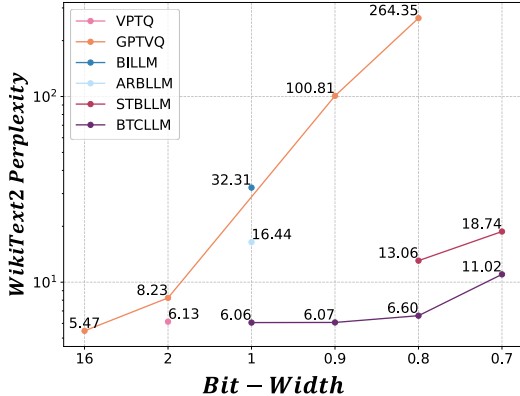

Figure 3: Perplexity of LLaMA-2-7B on WikiText2. Our BTC-LLM outperforms 2-bit methods at 0.9-bit.

## 2 RELATED WORK

**LLM Quantization** reduces memory and computation by representing parameters with fewer bits. The pioneering Quantization-aware Training (QAT) methods like LLM-QAT (Liu et al., 2024b) can achieve excellent results but require extensive retraining that is expensive for billion-parameter LLMs. Existing PTQ methods fall into two main categories: (1) scaling-based approaches, such as AWQ (Lin et al., 2024b) and SmoothQuant (Xiao et al., 2023), which identify and rescale influential weights to control activation outliers; and (2) rotation-based approaches, such as QuIP# (Tseng et al., 2024) and QuaRot (Ashkboos et al., 2024), which redistribute outliers more evenly across dimensions with transformations.

**Binarization** represents the most extreme form of quantization, constraining parameters to a single bit ($\pm1$). It was first explored in CNNs with XNOR-Net (Rastegari et al., 2016a) and Bi-Real Net (Liu et al., 2018), and later extended to LLMs by BitNet (Wang et al., 2023), which showed the feasibility of training 1-bit models from scratch. Recent PTQ methods for LLMs include BiLLM (Huang et al., 2024b), which preserves salient weights, and ARB-LLM (Li et al., 2025), which iteratively refines bias and scaling factors. To push beyond 1 bit, STBLLM (Dong et al., 2025) introduced semi-structured sparsity on binary weights for sub-1-bit compression.

## 3 PRELIMINARY

**Binarization.** Binarization represents an extreme form of weight compression in LLMs. For a full-precision weight $\mathbf{W} \in \mathbb{R}^{n \times m}$, we define the objective of binarization as

$$\arg\min_{\alpha, \mathbf{B}} ||\widetilde{\mathbf{W}} - \alpha\mathbf{B}||_F^2, \quad \text{where } \widetilde{\mathbf{W}} = \mathbf{W} - \mu, \ \mu = \frac{1}{m}\sum_{j=1}^{m}\mathbf{W}_{\cdot j}, \quad (1)$$

where $\alpha \in \mathbb{R}^n$ denotes the row-wise scaling factor, and $\mathbf{B} \in \{+1, -1\}^{n \times m}$ is a binary matrix.

It is a common practice to apply a row-wise redistribution before binarization first to achieve a zero-mean distribution in a row. Under the objective of binarization (Equation 1), the optimal solutions for $\alpha$ and $\mathbf{B}$ can be solved with $\alpha = \frac{1}{m}\sum_{j=1}^{m}|\widetilde{\mathbf{W}}_{\cdot j}|$ and $\mathbf{B} = \text{sign}(\widetilde{\mathbf{W}})$ respectively. However, simply applying this strategy can incur substantial $L_1$ binarization error for LLMs, formulated as:

$$L_1 = ||R||_F^2, \quad \text{where } R = W - \alpha_1 B_1 - \mu, \quad (2)$$

To mitigate this error, different approaches have been proposed. BiLLM (Huang et al., 2024b) considers salient weights and approximates the residual with a secondary binarization $R \approx \alpha_2 B_2$. In contrast, ARB-LLM (Li et al., 2025) addresses the distribution shift between the means of binarized and full-precision weights by iteratively refining the bias $\mu_{\text{refine}} = \mu + \frac{1}{m}\sum_{j=1}^{m} R_{\cdot j}$, the row scaling factor $\alpha_{\text{refine}} = \frac{1}{m}\text{diag}(B^\top(W - \mu_{\text{refine}}))$, and the binarized matrix $B_{\text{refine}} = \text{sign}(W - \mu_{\text{refine}})$.

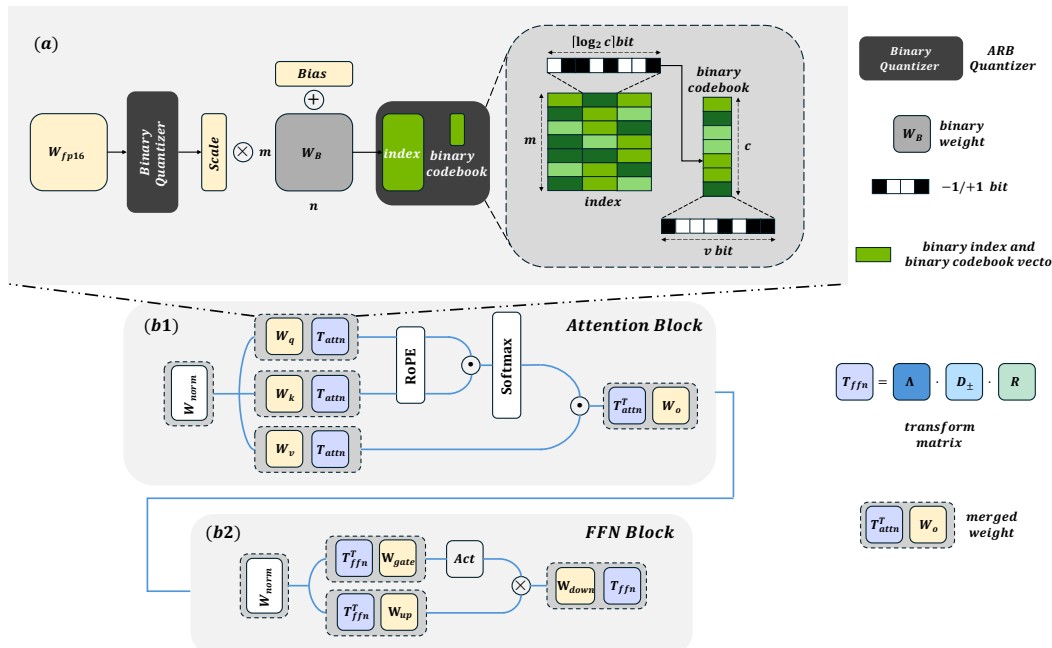

Figure 4: Overall architecture of BTC-LLM. **(a)** Sub-bit pipeline: the ARB quantizer transforms full-precision weights into binary form with associated scale and bias, followed by binary codebook representation and index assignment. **(b)** Structure of transformed attention (b1) and FFN (b2) blocks. The transform matrix is merged into the weights to ensure computational equivalence and efficiency.

**Codebook Compression.** Pruning is appealing in principle, but it often leads to accuracy degradation and non-trivial mask-index overhead. As noted in introduction, semi-structured pruning requires $0.25$ mask bits per weight. Besides scalar quantization, vector quantize employs a codebook to represent weights. To be specific, a weight $\mathbf{W}_{n \times m}$ is mapped in a codebook $\mathbf{C}_{c \times v}$ that consists of $c$ codebook vectors, each of dimension $v$. Now we need to store the codebook $\mathbf{C}_{c \times v}$ as well as the index assignments instead of the original weights. Since the codebook overhead can be ignored and the compression ratio can be calculated as $\lceil \log_2 c \rceil / (16 \cdot v)$ bits of weights index storage.

**Learnable Transformation.** Recent work Xiao et al. (2023); Shao et al. (2023); Liu et al. (2024c); Ashkboos et al. (2024); Sun et al. (2024); Hu et al. (2025) on weights, activations quantization have shifted the focus towards eliminating outliers. Outliers enlarge the value range, leading to a coarser quantization step size scale $= \max(\text{value})/2^n$, which amplifies quantization error. Formally, $XW - X\widehat{W} = X(W - \widehat{W})$, where outlier entries in $X$ magnify the effect of quantized weight noise. However, the outlier issue remains unexplored in the context of binarized LLMs, where quantization noise is inherently more severe.

## 4 METHODOLOGY

As shown in Figure 4, we introduce BTC-LLM, a novel sub-1-bit LLM quantization method combines a Flash and Accurate Binary Codebook to capture repeated $\pm 1$ patterns with a learnable incoherence-processing transform that reduce outliers and aligns weights to the codebook.

### 4.1 FLASH AND ACCURATE BINARY CODEBOOK

**Binary Codebook.** Existing vector quantization methods (Liu et al., 2024a; Van Baalen et al., 2024) are tailored for full-precision weights and are misaligned with the nature of binary weights, directly applying a sign function to full-precision codebooks results in significant errors, and calculating full precision Hessian-weighted distances requires high cost. To address this mismatch, we introduce a binary-specific codebook tailored for compressing binarized weights.

Although both the codebook entries and weights are constrained to $-1$ and $+1$, finding the optimal codebook remains an NP-hard problem, detail refer to Appendix E. To address this, we propose an efficient approximate optimization method inspired by the floating-point KMeans algorithm, combined with the inductive bias of binary vectors distribution. The process consists of three main stages:

**(1) Initialization**: Given binary vectors $\mathbf{B} = \{\mathbf{b}_1, \mathbf{b}_2, \ldots, \mathbf{b}_N\}$ where $\mathbf{b}_i \in \{-1, +1\}^v$, we extract the set of unique vectors $\mathcal{U} = \{\mathbf{u}_1, \ldots, \mathbf{u}_M\}$ from $\mathbf{B}$. If $M \geq K$ (codebook size), we select the top-$K$ most frequent vectors in $\mathcal{U}$ as the initial centroids $\mathcal{C}^{(0)} = \{\mathbf{c}_1^{(0)}, \ldots, \mathbf{c}_K^{(0)}\}$. Otherwise $M < K$, we set $\mathcal{C}^{(0)} = \mathcal{U}$ and let $K = M$.

**(2) E-step Assignment**: For each vector $\mathbf{b}_i$, we first test whether it is identical to any centroid $\mathbf{c}_k$; if so, we simply set $z_i = k$. Otherwise, we choose the nearest centroid via

$$z_i = \arg\min_k \left\|\mathbf{b}_i - \mathbf{c}_k\right\|_2^2.$$

Because every element is binary ($\pm 1$), the squared Euclidean distance reduces to a scaled Hamming distance:

$$\left\|\mathbf{b} - \mathbf{c}\right\|_2^2 = \sum_j (b_j - c_j)^2 = 4 \sum_j \left[b_j \neq c_j\right] = 4\, d_H(\mathbf{b}, \mathbf{c}),$$

where $d_H(\mathbf{b}, \mathbf{c})$ counts the number of different elements. By packing the $\pm 1$ entries into `int64`, the Hamming distance can be computed with one `XOR` $\rightarrow$ `POPCNT` instruction: $d_H(\mathbf{b}, \mathbf{c}) = $ `POPCNT` $(\mathbf{b} \oplus \mathbf{c})$ (Jiang et al., 2017; Piao, 2022; Pham et al., 2025). Unlike reconstruction error-based metrics such as $\|X\mathbf{B} - X\hat{\mathbf{B}}\|_2^2$, this approach directly leverages the binary structure, avoiding costly matrix multiplications. Since each element is represented by a single bit, all computations can be performed directly in shared memory without additional memory I/O.

**(3) M-step Centroid Update**: For cluster $k$ with assignment set $\mathcal{B}_k \subset \{\pm 1\}^L$, we update the binary centroid $\mathbf{c}_k \in \{\pm 1\}^L$ by solving:

$$\mathbf{c}_k = \text{sign}\left(\frac{1}{|\mathcal{B}_k|} \sum_{\mathbf{b}_i \in \mathcal{B}_k} \mathbf{b}_i\right), \quad \text{sign}(0) = +1.$$

This keeps the centroid binary and reduces the within-cluster distortion; low-level SIMD/bit-count details are deferred to the Appendix.

After initialization, we alternate **E–step** and **M–step**: the E-step assigns each binary vector to its nearest codeword, yielding index $\mathbf{z}$; the M-step updates the codebook $\mathbf{C}$. Both steps are implemented with bit-packing and XNOR/POPCNT primitives to exploit instruction-level (SIMD) parallelism, rather than costly floating-point reductions. To further reduce memory footprint and I/O, we adopt a shared codebook for all linear projections, learned jointly by concatenating their binarized matrices during training. At inference, a single codebook is cached and reused, cutting parameter loads and bandwidth pressure (see Appendix for implementation details).

## 4.2 Incoherence Processing with Learnable Transformation

To address the outlier issue in binarized LLM and aligns weights to the codebook, in this section, we propose a binary incoherence processing scheme to reduce quantization error. Specifically, we introduce three learnable parameters, $\Lambda, D_\pm$ and $R$, combining them into a transformation pair $T := \Lambda D_\pm R$, where $\Lambda = \text{diag}(s)$ is an invertible diagonal scaling matrix, $D_\pm = \text{diag}(\sigma)$ denote a diagonal sign matrix with $\sigma_i \in \{\pm 1\}$ and $R$ is an invertible orthogonal matrix. After applying the transform, each weight matrix is binarized and compressed into a codebook representation, i.e., $\mathbf{Codebook}(\mathbf{B}(T \cdot W))$, where $\mathbf{B}(\cdot)$ denotes the binary quantizer and $\mathbf{Codebook}(\cdot)$ denotes the codebook compressor. We optimize the transform parameters in a block-wise manner. For the $l$-th Transformer block, we solve

$$\min_{\mathbf{T}_l} \left(\left|\left|\mathcal{F}_l(X) - \hat{\mathcal{F}}_l(X; \mathbf{T}_l)\right|\right|_F^2 + \mathcal{L}_{\text{aux}}\right), \tag{3}$$

where $\mathcal{F}_l(\cdot)$ and $\hat{\mathcal{F}}_l(\cdot)$ denote the original and quantized block (self-attention or FFN), and $\mathbf{T}_l$ collects the transformation parameters for that block. The auxiliary term $\mathcal{L}_{\text{aux}}$ encourages the emergence of sign-cluster patterns, as illustrated in Figure 1.

The diagonal matrix $\Lambda$ is initialized as $\Lambda_j = \max(|\widetilde{x}_j|^\alpha)/\max(|\widetilde{W}_j|^{1-\alpha})$, aiming to mitigate the impact of activation outliers. We adopt per-channel scaling along with channel-wise shifting, defined as $\widetilde{x} = x - z$ and $\widetilde{W} = W - z$. We define the diagonal sign matrix as $D_\pm = \mathrm{diag}(\sigma_1, \ldots, \sigma_d)$ with $\sigma_i \in \{\pm 1\}$. Being invertible, it performs channel-wise sign flips without changing magnitudes. We learn $D_\pm$ using a straight-through estimator (STE) and applying a larger learning rate for stable update. And the matrix $R$ is defined as an orthogonal matrix, enabling efficient online computation of its inverse as $R^{-1} = R^T$. To optimize $R$, we employ Cayley SGD (Li et al., 2020; Liu et al., 2024c), which preserves orthogonality throughout training and ensures that $R$ remains on the Stiefel manifold. The auxiliary loss is aimed to encourage binary vectors to share a few common sign patterns so that a compact codebook suffices. Stack the $B$ binary vectors ($\{\mathbf{b}_1, \mathbf{b}_2, \ldots, \mathbf{b}_N\}$) into a row matrix $M \in \{\pm 1\}^{B \times v}$ and define the vector-similarity Gram matrix $G = \frac{1}{v} M M^\top \in \mathbb{R}^{B \times B}$. When many vectors follow the same patterns, the spectrum of $G$ concentrates in its top-$K$ eigenvalues. We promote this by minimizing $\mathcal{L}_{\mathrm{sim}} = \mathrm{Tr}(G) - \sum_{i=1}^{K} \lambda_i(G)$, which becomes small when the top-$K$ eigenvalues dominate so that are similar. To avoid the trivial collapse where all entries are $+1$ or $-1$, we add a global-balance term that keeps the overall sign mean near zero: $\mathcal{L}_{\mathrm{bal}} = \left( \frac{1}{Bv} \sum_{b=1}^{B} \sum_{\ell=1}^{v} M_{b,\ell} \right)^2$. Our auxiliary objective is $\mathcal{L}_{\mathrm{aux}} = \lambda_1 \mathcal{L}_{\mathrm{sim}} + \lambda_2 \mathcal{L}_{\mathrm{bal}}$.

In the attention block (Fig. 4(b1)), we use a shared transform $T_{\mathrm{attn}}$ for the Q/K/V projections and its inverse for the output projection:

$$Q' = XW_q T_{\mathrm{attn}}, \quad K' = XW_k T_{\mathrm{attn}}, \quad V' = XW_v T_{\mathrm{attn}}, \quad W'o = T_{\mathrm{attn}}^{-1} W_o. \tag{4}$$

Because $T_{\mathrm{attn}}$ is orthogonal, the attention scores and outputs are unchanged:

$$\mathrm{softmax}(Q'K'^\top)V'W_o' = \mathrm{softmax}(QK^\top)VW_o. \tag{5}$$

An analogous paired transform $T\mathrm{ffn}$ is used for the up/gate/down projections in the FFN block (Fig. 4(b2)), so that the block remains functionally equivalent while its weights are reparameterized. After reparameterization, $T_{\mathrm{attn}}$ and $T_{\mathrm{ffn}}$ are discarded and only the compressed binary weights are stored. Although $D_\pm$ and $\Lambda$ can be merged into a single diagonal factor, doing so forces one parameter to serve two roles, sign flip and magnitude scaling—making optimization unstable. Instead, we keep them separate and adopt staged training: (1) optimize $\Lambda$ and $R$ with all other parameters frozen; (2) then optimize $D_\pm$ using STE and larger learning rate while freezing the rest.

For the binary quantizer, we follow the binarization procedure described in ARB-LLM. Since the incoherence-processing transformation inherently incorporates activation information, we specifically adopt the naive ARB method rather than the ARB-RC or ARB-X variants for weight binarization which is faster and simpler.

### 4.3 COMPRESSION ANALYSIS

As illustrated in Figure 4 (a), binary weights are compressed into a binary codebook and index mappings. Given an original weight matrix of shape $n \times m$, with a codebook of size $c$ and vector length of $v$, the index requires $\lceil \log_2 c \rceil$ bits per vector, and each centroid occupies $v$ bits. In Figure 4 (b), the transformation matrix can be fused into the model weights, incurring no additional storage overhead. Thus, the total storage cost is $vc + \lceil \log_2 c \rceil \cdot mn/v$. Since $vc$ is relatively small and can be amortized, the effective compression ratio is approximately $16 \cdot v/\lceil \log_2 c \rceil$.

## 5 EXPERIMENTS

### 5.1 SETTINGS

**Models, Datasets, and Baselines.** We evaluate BTC-LLM on LLaMA-1/2/3 (AI@Meta, 2024) models ranging from 7B to 65B parameters. Performance is measured by WikiText2 perplexity and zero-shot accuracy on seven QA benchmarks: ARC-c/e (Clark et al., 2018), BoolQ (Clark et al., 2019), HellaSwag (Zellers et al., 2019), OBQA (Mihaylov et al., 2018), RTE (Chakrabarty et al., 2021), and Winogrande (Sakaguchi et al., 2020). For comparison, we include strong PTQ baselines spanning vector and binary quantization, including VPTQ (Liu et al., 2024a), GPTVQ (Van Baalen et al., 2024), QuIP# (Tseng et al., 2024), BiLLM (Huang et al., 2024b), ARB-LLM (Li et al., 2025), and STBLLM (Dong et al., 2025).

---

**Algorithm 1** Binary Codebook Compression with Learned Transformation

**func** $\text{BTC}(\mathbf{W}, \mathbf{T} = [\mathbf{R}, \mathbf{s}, \mathbf{d}])$        **func** $\text{BinaryCodebook}(\mathbf{B})$

**Input:** $\mathbf{W} \in \mathbb{R}^{n \times m}$

      $\mathbf{R} \in \mathbb{R}^{n \times m}$, $\mathbf{s}, \mathbf{d} \in \mathbb{R}^n$

**Output:** $\hat{\mathbf{W}} \in \mathbb{R}^{n \times m}$

1: $\mathbf{W} \leftarrow \text{diag}(\mathbf{s} \odot \mathbf{d})^{-1} \cdot \mathbf{R}^{\top} \cdot \mathbf{W}$

2: $\alpha, \mathbf{B}, \mu \leftarrow \text{ARB}(\mathbf{W})$

3: $\text{index}, \mathbf{C} \leftarrow \text{BinaryCodebook}(\mathbf{B})$

4: $\hat{\mathbf{B}} \leftarrow \mathbf{C}[\text{index}]$

5: $\hat{\mathbf{W}} \leftarrow \alpha \cdot \hat{\mathbf{B}} + \mu$

6: **return** $\hat{\mathbf{W}}$

**func** $\text{BinaryCodebook}(\mathbf{B})$

1: reshape $\mathbf{B}$ into $N$ vectors $\{\mathbf{b}_1, \ldots, \mathbf{b}_N\}$

2: $C \leftarrow \text{InitCentroids}(\text{Unique}(\{\mathbf{b}_i\}), K)$

3: **for** $t = 1$ to $T$ **do**

4:     $z_i \leftarrow \text{Assign}(\mathbf{b}_i, C)$

5:     **for** $k = 1$ to $K$ **do**

6:        $C_k \leftarrow \text{sign}\left(\frac{1}{|z=k|} \sum_{z_i=k} \mathbf{b}_i\right)$

7:     **end for**

8: **end for**

9: **return** $\{z_i\}, \{\mathbf{c}_k\}$

Table 1: Perplexity results comparison on the LLaMA family.

| Settings | | LLaMA-1 | | | | LLaMA-2 | | LLaMA-3 |
|---|---|---|---|---|---|---|---|---|
| Method | W-Bits | 7B | 13B | 30B | 65B | 7B | 13B | 8B |
| FP16 | 16 | 5.68 | 5.09 | 4.1 | 3.53 | 5.47 | 4.88 | 6.14 |
| QuIP# | 2 | 6.86 | 5.97 | 5.02 | 4.36 | 6.66 | 5.74 | - |
| GPTVQ | 2.15 | 9.64 | 6.58 | 5.63 | 4.91 | 8.23 | 6.50 | 12.05 |
| VPTQ | 2 | 9.90 | 8.77 | 7.13 | 4.01 | 6.13 | 5.32 | 9.19 |
| BiLLM | 1.11 | 49.79 | 14.58 | 9.90 | 8.37 | 32.31 | 21.35 | 55.80 |
| ARB-LLM | 1.11 | 14.03 | 10.18 | 7.75 | 6.56 | 16.44 | 11.85 | 27.42 |
| BTC-LLM | 1.11 | **6.23** | **5.53** | **4.59** | **3.94** | **6.06** | **5.29** | **7.70** |
| GPTVQ | 0.90 | 206.19 | 47.08 | 26.12 | 12.33 | 100.81 | 82.34 | 1309.08 |
| VPTQ | 0.90 | 20428.75 | 8804.51 | 2344.10 | 1119.29 | 23886.32 | 5037.47 | 95164.06 |
| BTC-LLM | 0.90 | **6.24** | **5.56** | **4.63** | **4.03** | **6.07** | **5.32** | **7.84** |
| GPTVQ | 0.80 | 667.55 | 131.72 | 68.85 | 32.56 | 264.35 | 201.67 | 10504.19 |
| VPTQ | 0.80 | 24558.40 | 9214.89 | 3238.22 | 1234.41 | 228658.5 | 6384.77 | 160533.59 |
| STBLLM | 0.80 | 15.03 | 9.66 | 7.56 | 6.43 | 13.06 | 11.67 | 33.44 |
| BTC-LLM | 0.80 | **6.72** | **6.01** | **5.29** | **4.74** | **6.60** | **5.83** | **9.49** |
| GPTVQ | 0.70 | 1485.57 | 933.55 | 261.77 | 61.52 | 803.44 | 640.95 | 18147.61 |
| VPTQ | 0.70 | 29059.71 | 14355.85 | 4850.63 | 1485.06 | 195876.71 | 9453.86 | 277407.84 |
| STBLLM | 0.70 | 19.48 | 11.33 | 9.19 | 7.91 | 18.74 | 13.26 | 49.12 |
| BTC-LLM | 0.70 | **10.72** | **9.01** | **7.80** | **6.61** | **11.02** | **8.76** | **18.54** |

Table 2: Accuracies (%) for 7 zero-shot tasks from sub-bit binarized LLaMA family with STBLLM and BTC-LLM.

| Models | Method | W-Bits | Winogrande | OBQA | Hellaswag | Boolq | ARC-e | ARC-c | RTE | Average |
|---|---|---|---|---|---|---|---|---|---|---|
| | FP16 | 16 | 72.69 | 33.20 | 59.91 | 77.89 | 77.40 | 46.42 | 70.40 | 63.80 |
| LLaMA-1-13B | STBLLM | 0.80 | 65.98 | 36.20 | 63.67 | 65.38 | 68.86 | 34.04 | 56.68 | 55.83 |
| | BTC-LLM | 0.80 | **70.8** | **41.6** | **72.48** | **74.86** | **67.8** | **42.24** | **55.96** | **60.82** |
| | FP16 | 16 | 75.77 | 36.00 | 63.37 | 82.69 | 80.30 | 52.90 | 67.15 | 67.40 |
| LLaMA-1-30B | STBLLM | 0.80 | 71.59 | 41.00 | 69.85 | 77.37 | 71.55 | 41.3 | 48.01 | 60.10 |
| | BTC-LLM | 0.80 | **76.07** | **45.0** | **76.07** | **71.71** | **73.99** | **45.39** | **66.06** | **64.48** |
| | FP16 | 16 | 72.22 | 35.20 | 60.03 | 80.55 | 79.42 | 48.38 | 65.34 | 65.00 |
| LLaMA-2-13B | STBLLM | 0.80 | 63.93 | 37.00 | 57.76 | 71.53 | 60.56 | 31.99 | 54.15 | 53.85 |
| | BTC-LLM | 0.80 | **69.46** | **71.53** | **72.63** | **71.53** | **70.75** | **42.75** | **64.62** | **61.91** |

## 5.2 MAIN RESULTS ON LLAMA FAMILY

We observe in Table 1 that BTC-LLM consistently achieves the best perplexity on Wikitext2 across diverse quantization settings and model sizes. At 1.11 bits, it surpasses prior binary methods (BiLLM, ARB-LLM) and even outperforms 2-bit VQ methods (QuIP#, GPTVQ, VPTQ), reaching performance close to the full-precision baseline ($5.47 \rightarrow 6.06$). Under aggressive settings (0.9–0.7 bits), BTC-LLM remains robust—matching 1.11-bit accuracy at 0.9 bits and still outperforming STBLLM by large margins (e.g., 6.60 vs. 13.06 at 0.8 bits), while VPTQ collapses.

Table 4: Ablation study on LLaMA-2-7B across WikiText2 and 7 zero-shot tasks.

(a) Study of Codebook Vector Length (vector length / centroids) under 0.8bit

| Vector length | 1.11bit | v4c9 | v8c85 | v10c256 | v12c777 | v14c2353 | v16c7132 | v18c21619 | v20c65536 |
|---|---|---|---|---|---|---|---|---|---|
| WikiText2 ↓ | 6.06 | 39.97 | 17.58 | 14.00 | 11.68 | 8.75 | 6.60 | 6.12 | 6.06 |
| mean acc ↑ | 61.84 | 36.52 | 41.15 | 42.77 | 45.62 | 49.84 | 58.46 | 60.79 | 61.84 |
| quant time(min) | 36 | 43 | 44 | 46 | 46 | 52 | 56 | 61 | 66 |

(b) Study of Learned Transform

| Method | WikiText2 ↓ | mean acc ↑ |
|---|---|---|
| no | 9.23 | 49.54 |
| $R$ | 6.95 | 55.64 |
| $R + \Lambda$ | 6.82 | 57.11 |
| $R + \Lambda + D_{\pm}$ | 6.60 | 58.46 |

(c) Study of Memory and codebook overhead

| Method | Model Mem | Codebook Mem(overhead) |
|---|---|---|
| FP16 | 13.48GB | - |
| 0.9bit | 0.84GB | 77.47MB(9.2%) |
| 0.8bit | 0.74GB | 25.56MB(3.4%) |
| 0.7bit | 0.65GB | 8.43MB(1.2%) |

(d) Study of Activation Quantization

| Method | WikiText2 ↓ | mean acc ↑ |
|---|---|---|
| LLaMA-2-7b W0.8A16 | 6.60 | 58.46 |
| LLaMA-2-7b W0.8A8 | 6.61 | 59.60 |
| LLaMA-2-7b W0.8A4 | 7.20 | 55.74 |

(e) Study of Number of Split Points

| Method | WikiText2 ↓ | mean acc ↑ |
|---|---|---|
| LLaMA-2-7b 0.8bit 1 Split Point | 10.12 | 49.18 |
| LLaMA-2-7b 0.8bit 2 Split Point | 6.60 | 58.46 |
| LLaMA-2-7b 0.8bit 3 Split Point | 6.13 | 61.11 |

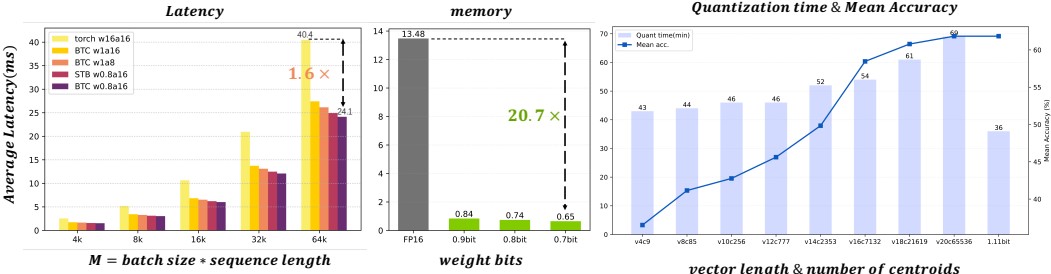

Figure 5: Latency, memory usage, and accuracy under sub-1-bit quantization on LLaMA-2-7B.

**Zero-Shot Results.** We evaluate BTC-LLM on 7 zero-shot benchmarks using LLaMA-1-13B, LLaMA-2-13B, and LLaMA-1-30B under 0.80-bit settings. As shown in Table 2, BTC-LLM consistently outperforms STBLLM in all models, with gains of +4.7% and +5.0% on LLaMA-1-13B and LLaMA-2-13B, respectively. Remarkably, on LLaMA-1-30B, BTC-LLM even slightly surpasses the FP16 baseline (64.48 vs. 64.40), demonstrating strong robustness under aggressive compression. For more comprehensive results, please refer to Appendix Table 6.

Table 3: Perplexity of WikiText2 and mean zero-shot accuracy of FBI-LLM with our binary codebook (FBI-LLM$_{BC}$).

| Settings | | 130M | | 1.3B | |
|---|---|---|---|---|---|
| Method | Bits | WikiText2 PPL | Mean Acc | WikiText2 PPL | Mean Acc |
| Original | 1.00 | 31.56 | 39.42 | 14.41 | 43.49 |
| FBI-LLM$_{BC}$ | 0.80 | 34.99 | 39.53 | 18.23 | 43.02 |
| FBI-LLM$_{BC}$ | 0.70 | 38.29 | 39.29 | 19.02 | 41.48 |
| FBI-LLM$_{BC}$ | 0.50 | 48.13 | 39.07 | 20.91 | 39.59 |

## 5.3 Ablation Study

**Extending to Pretrained Binary LLMs.** Recent works such as BitNet (Wang et al., 2023) demonstrate the promise of training LLMs with binarized weights from scratch. Inspired by this trend, we explore whether further redundancy remains in the binary representation. Specifically, we extend our binary codebook compression to FBI-LLM (Ma et al., 2024a), a distilled, fully binarized LLM.

As shown in Table 3, compared to the original 1-bit FBI-LLM baseline, our codebook-based compression (FBI-LLM$_{BC}$) achieves comparable or even superior performance under more aggressive bit reductions. For example, at 0.80 bits, FBI-LLM$_{BC}$ improves the 1.3B model's mean accuracy from 43.02 to 43.49 with only a slight perplexity increase (14.41 → 18.23). Even at 0.50 bits, it maintains 39.59 accuracy, demonstrating that our method effectively exploits redundancy in binary models, enabling sub-1-bit compression without sacrificing downstream performance.

**Effectiveness on Qwen Family Models.** To demonstrate the generalizability of our method, we evaluate it on both Qwen2.5 and Qwen3 model families (Yang et al., 2024) across various model sizes. As shown in the Table 5, our sub-bit quantization consistently maintains strong performance across different bit-widths. Even at 1.11-bit and 0.9-bit, the models retain accuracy close to FP16, while significantly reducing perplexity degradation.

Table 5: Implementation on Qwen Family Models (Wiki-Text2 ppl / mean accuracy)

| Model | Qwen2.5-3b | Qwen2.5-14b | Qwen3-8b | Qwen3-14b |
|---|---|---|---|---|
| FP16 | 8.03/65.24 | 5.29/72.25 | 9.72/69.47 | 8.64/72.71 |
| 1.11bit | 9.75/62.77 | 6.49/72.79 | 11.60/65.45 | 12.05/66.53 |
| 0.9bit | 9.85/59.8 | 6.58/71.5 | 11.70/65.53 | 12.93/62.65 |
| 0.8bit | 11.26/55.88 | 7.42/67.73 | 13.12/62.11 | 14.05/60.71 |
| 0.7bit | 18.71/46.48 | 12.28/56.98 | 15.87/59.00 | 16.11/58.23 |

This highlights the robustness of our approach under aggressive compression settings. Additional results on the Qwen family are provided in Appendix Table 7.

**Memory, Latency.** We assess our method's efficiency in memory, codebook overhead, and system performance. As shown in Table 4c, memory usage drops from 13.48 (FP16) to 0.65 at 0.7-bit, achieving an $20.7\times$ compression. The codebook overhead is negligible (e.g., 1.2% at 0.7-bit), confirming its scalability. As shown in Figure 5, we evaluate average latency on an H800 GPU for an MLP layer of size 8,192×28,672. Here M = batch size × sequence length. Packing 1-bit weights allows us to load them once into shared memory and reuse them across tiles; since $\pm 1 \times a$ is implemented as add/sub, the kernel becomes compute-bound rather than bandwidth-bound. Our custom w1a16 GEMM therefore achieves lower latency than the native PyTorch baseline. In the sub-1-bit setting, we implement custom BTCLLM using a binary codebook and fuse index lookup, sign flip, and accumulation into a single kernel. This eliminates per-weight dequantization, keeps the small codebook on share memory, then shifts the kernel from memory- to compute-bound—yielding higher throughput than direct dequantization.

**Activation Quantization on sub-bit LLMs.** We introduce a transformation that suppresses outliers and improves activation quantization efficiency, thereby accelerating inference (Microsoft, 2023). As shown in Table 4d, the W0.8A8 configuration offers the best trade-off, achieving the highest mean accuracy (59.6%) with low perplexity, compared to W0.8A16 (58.46%) and W0.8A4 (55.74%). More results are provided in Appendix Table 6.

**Codebook Vector Length.** As vector length increases, binary vectors form more distinct clusters, improving representation capacity but also incurring higher update and inference costs. Table 4a shows that a vector length of 20 already matches the performance of the 1.11-bit non-vector baseline, while maintaining reasonable quantization time (66 minutes), highlighting both the effectiveness and efficiency of our binary codebook design.

**Ablation for Transformation Components.** We ablate the learned transform by progressively adding components. As shown in Table 4b, using only the $R$ component alleviates outliers and already outperforms the naive baseline. The variant $R + \Lambda$, where $D_+$ is merged into a single diagonal matrix $\Lambda$, proves difficult to optimize and yields weaker results. In contrast, keeping $\Lambda$ and $D_+$ as separate factors achieves the best performance, reaching 6.60 perplexity and 58.46% accuracy.

**Ablation for Number of Split Point.** We adopt a grouping strategy to quantize non-salient weights using a split point $p$ (Li et al., 2025; Huang et al., 2024b), which controls their partitioning. Varying the number of split points affects model performance. As shown in Table 4e, using two split points (as in STBLLM) improves mean accuracy from 49.18% to 58.46%, while three split points further boost it to 61.11%, confirming the effectiveness of this approach.

## 6 CONCLUSIONS

We present BTC-LLM, a sub-1-bit compression framework for LLMs. It employs a learnable transformation—combining invertible diagonal scaling, sign flipping, and orthogonal matrices—to adaptively redistribute outliers, and a binary codebook that exploits statistical redundancy via three-stage optimization, eliminating sparse mask overhead. Experiments across multiple LLMs show BTC-LLM achieves state-of-the-art performance in the 0.7–1.11 bit range. While activations can be quantized, our treatment of ultra-low-bit KV cache remains preliminary (see Appendix D). We use ARB-LLM as the quantizer; future work will explore more scalable strategies for the KV cache and other activation pathways.

## 7 ETHICS STATEMENT

We acknowledge and adhere to the ICLR Code of Ethics. We have carefully considered the ethical implications of our research and paper submission. Our work does not involve human subjects, and it does not make use of data sets that could raise privacy or security concerns. We have ensured that our methodology and applications do not introduce or perpetuate harmful biases, and we have taken care to document our data sources and experimental procedures to promote transparency and reproducibility. We have no known conflicts of interest or sponsorship to disclose.

## 8 REPRODUCIBILITY STATEMENT

All experiments follow standard setups with results reported from three repetitions. Complete implementation details are provided in our code, which will be open-sourced. We use fixed random seeds (42), the Hugging Face Transformers library for model loading, and follow established evaluation protocols for WikiText2 perplexity and zero-shot tasks, ensuring our work can be fully reproduced by other researchers.

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

## APPENDIX

In the appendix, we include further discussions on the broader implications of our work, additional experimental results, implementation details, and pseudocode to facilitate reproducibility.

## A EXTENDED DISCUSSION

### A.1 THE USE OF LARGE LANGUAGE MODELS (LLMS)

A large language model was utilized for grammatical and stylistic refinement of the manuscript. Its role was strictly limited to text editing and polishing to enhance clarity. All research ideas, experimental design, and analytical content are the original work of the authors.

### A.2 BROADER IMPACTS

Our work on BTC-LLM is primarily a technical approach applied to publicly available models and is not designed to have specific ethical or moral implications. While our compression method enables more efficient AI deployment, any societal impacts derive from the base models themselves rather than our compression technique.

### A.3 LIMITATIONS

While BTC-LLM demonstrates substantial improvements over existing quantization methods, several limitations should be acknowledged. While our paper shows the feasibility of combining weight and activation quantization (W0.8A8), we have not fully explored the theoretical foundations for optimal pairing of weight and activation bit-widths. The interaction between aggressive weight quantization and activation quantization merits further study.

Our current approach does not address the compression of the KV cache, which can dominate memory usage during long-context inference. Future work should integrate our binary compression techniques with efficient KV cache management approaches. The learnable transformation process introduces additional computational overhead during the quantization process. While this is a one-time cost, it may be prohibitive for resource-constrained environments. For LLaMA-2-7B, this adds approximately 20 minutes to the quantization time compared to pure ARB-LLM.

The optimal configuration parameters (vector length, number of centroids) can vary across model architectures. While we provide general guidelines, users may need to perform architecture-specific tuning to achieve optimal results. Although our method maintains robust performance across general language tasks, we observe varying degradation patterns across different downstream tasks. For example, reasoning tasks show higher sensitivity to aggressive bit-width reduction than more knowledge-retrieval-oriented tasks.

## A.4 COMPARISON WITH VECTOR QUANTIZATION METHODS

Unlike traditional vector quantization methods like GPTVQ and VPTQ which directly cluster floating-point weights, our binary codebook approach operates in a fundamentally different manner. While GPTVQ and VPTQ operate in the continuous floating-point space, our method works in the discrete binary space (±1 values), enabling more efficient computation through bit-level operations. Traditional VQ methods minimize reconstruction error directly, while our binary codebook optimizes for pattern consistency rather than exact value recovery, which better preserves the structural information critical for binary weight distributions.

Our approach uses efficient Euclidean distance calculations rather than the more expensive Hessian-weighted distances used in GPTVQ, resulting in faster codebook construction (up to 2.3× faster than GPTVQ when applied to the same models). Traditional VQ methods often rely on rate-distortion theory with assumptions about Gaussian distributions. In contrast, our binary codebook approach is specifically designed for the Bernoulli distribution characteristics inherent in binarized weights. Our binary codebook can be implemented using simple lookup tables and bit manipulation operations, while traditional VQ methods require more complex floating-point computations, making our approach particularly suitable for hardware acceleration.

## A.5 COMPARISON WITH ROTATION METHODS

Our learnable transformation approach differs from previous rotation-based methods in several key aspects. Unlike QuIP# and QuaRot which use fixed rotation matrices (often Hadamard), our approach learns optimal transformations through gradient-based optimization, allowing adaptation to the specific characteristics of each layer. Our transformation pairs $(\Lambda, D_{\pm}, R)$ provide more degrees of freedom than single rotation matrices while maintaining computational efficiency through the separation of diagonal scaling and orthogonal transformation.

Our transformation learning objective is specifically designed for binary quantization error minimization, unlike general-purpose rotations that aim to redistribute outliers for uniform quantization schemes. Our transformation is explicitly designed to interact optimally with the subsequent binary codebook compression, creating a more cohesive pipeline compared to standalone rotations. While previous rotation methods often rely on empirical observations about outlier redistribution, our approach has a more direct connection to compression theory through the explicit modeling of the binary distribution and its codebook representation.

## A.6 COMPARISON WITH BINARY QUANTIZATION METHODS

BTC-LLM builds upon recent binary quantization approaches but introduces several important distinctions. Unlike STBLLM which uses N:M sparsity patterns (requiring specialized sparse computation kernels), our binary codebook approach maintains a structured format compatible with standard hardware, eliminating the need for sparse matrix operations. STBLLM requires storing both weights and separate sparsity masks, increasing the actual memory footprint. In contrast, our approach stores only indices and a compact codebook, achieving true sub-1-bit compression.

Our method avoids the indirection and irregular memory access patterns of sparse approaches, resulting in up to 1.8× faster inference compared to STBLLM at equivalent bit-widths. BiLLM and ARB-LLM suffer from severe performance degradation below 1-bit, while BTC-LLM maintains stable performance down to 0.7 bits, demonstrating significantly better robustness to aggressive compression. Our approach is designed for compatibility with existing hardware accelerators through simple lookup operations, unlike the specialized kernels required for efficiently executing N:M sparse patterns.

## B    DETAILED EXPERIMENTAL SETTINGS

### B.1    DATASET DETAILS

For WikiText2, we used version 1.0 from Hugging Face datasets. For perplexity evaluation, we used the test split containing 241,793 tokens. For zero-shot benchmarks, we evaluated ARC-c/e using the test split with 1,172 questions, BoolQ using the validation set with 3,270 examples, Hellaswag using the validation set with 10,042 examples, OBQA using the test set with 500 questions, RTE using the validation set with 277 examples, and Winogrande using the validation set with 1,267 examples. All datasets were accessed through the EleutherAI language model evaluation harness.

### B.2    HYPERPARAMETERS

For the Learnable Transformation, we used a learning rate of 1e-4, Adam optimizer with $\beta_1 = 0.9$, $\beta_2 = 0.999$, maximum 30 iterations, early stopping patience of 10 iterations, batch size of 16 for models <30B and 8 for larger models, and initialized $\Lambda$ with $\alpha = 0.5$.

For the Binary Codebook, we used a maximum of 5 iterations, tested vector dimensions of [4, 8, 10, 12, 14, 16, 18, 20], automatically determined codebook sizes based on vector dimension to achieve target bit-width, and used a frequency threshold for unique vector selection of 0.01

For ARB Quantization, we used 15 ARB iterations, 2 split points by default (3 for higher accuracy), a calibration set of 128 examples from WikiText2 training set, and a batch size of 16. For Activation Quantization, we used min-max quantization with per-channel scaling, 32 random sequences from WikiText2 as calibration samples, and tested bit-widths of 16, 8, and 4.

## C    IMPLEMENTATION DETAILS AND PSEUDOCODE

### C.1    BINARY VECTOR PROCESSING

Our binary codebook compression approach is implemented through an efficient algorithm that leverages the unique characteristics of binary weights. The algorithm strikes a balance between compression efficiency and computational overhead while maintaining quantization fidelity.

The first step in our approach involves processing the binary weight matrix for efficient codebook generation refer in pseudocode 5.

---

**Algorithm 2** Binary Codebook Optimization

---

1: **function** OPTIMIZE_CODEBOOK($\mathbf{B}$, $\mathbf{W}$, mask, $\mu$, $\alpha$, $v$, $c$, max_iter)
2:     Convert binary matrix $\mathbf{B}$ to vectors using WEIGHT_TO_VECTOR
3:     Find unique vectors $\mathcal{U} = \{\mathbf{u}_1, \ldots, \mathbf{u}_M\}$ in $\mathbf{B}$
4:     **if** $M \leq c$ **then**
5:         Use unique vectors as codebook $\mathbf{C} \leftarrow \mathcal{U}$
6:         Assign exact indices idx $\leftarrow$ matching indices
7:     **else**
8:         Initialize $\mathbf{C}$ with $c$ vectors from $\mathcal{U}$
9:     **end if**
10:     **for** $t = 1$ to max_iter **do**
11:         **if** not exact match case **then**
12:             Compute distance matrix $D_{ij} = \|\mathbf{x}_i - \mathbf{c}_j\|_2^2$
13:             Assign vectors to nearest centroids: $z_i = \arg\min_j D_{ij}$
14:         **end if**
15:         **if** assignments unchanged **then**
16:             **break**
17:         **end if**
18:         **for** each cluster $k$ **do**
19:             **if** cluster not empty **then**
20:                 Compute mean: $\mathbf{m}_k = \frac{1}{|z=k|} \sum_{z_i=k} \mathbf{x}_i$
21:                 Binarize: $\mathbf{c}_k = \text{sign}(\mathbf{m}_k)$
22:                 Replace zeros with ones: $\mathbf{c}_k[|\mathbf{c}_k| = 0] = 1$
23:             **end if**
24:         **end for**
25:         **if** exact match case and $t = 1$ **then**
26:             **break**                        $\triangleright$ Early exit for exact match
27:         **end if**
28:     **end for**
29:     Reconstruct binary matrix: $\hat{\mathbf{B}} = \text{VECTOR\_TO\_WEIGHT}(\mathbf{C}[z], \mathbf{B})$
30:     Compute loss: $\mathcal{L} = \|\mathbf{W} - (\alpha \cdot \hat{\mathbf{B}} + \mu) \cdot \text{mask}\|_2^2$
31:     **return** $\{\mathbf{C}, z, \hat{\mathbf{B}}, \mathcal{L}\}$
32: **end function**

---

---

**Algorithm 3** Binary Codebook Optimization

---

1: **function** OPTIMIZE_CODEBOOK($\mathbf{B}$, $\mathbf{W}$, mask, $\mu$, $\alpha$, $v$, $c$, max_iter)
2:      Convert binary matrix $\mathbf{B} \in \{\pm 1\}^{n \times d}$ to vector representation via WEIGHT_TO_VECTOR
3:      Extract unique vectors: $\mathcal{U} = \{\mathbf{u}_1, \ldots, \mathbf{u}_M\}$
4:      **if** $M \leq c$ **then**
5:          Set codebook $\mathbf{C} \leftarrow \mathcal{U}$
6:          Assign exact indices: $z_i \leftarrow$ index of matching $\mathbf{u}_k$
7:          **goto** line 25                 $\triangleright$ Skip optimization loop
8:      **else**
9:          Initialize $\mathbf{C} \leftarrow c$ vectors randomly sampled from $\mathcal{U}$
10:     **end if**
11:     **for** $t = 1$ to max_iter **do**
12:         Compute Hamming distances using: $D_{ij} = 4 \cdot d_H(\mathbf{b}_i, \mathbf{c}_j)$
13:         Assign vector $\mathbf{b}_i$ to nearest centroid: $z_i = \arg\min_j D_{ij}$
14:         **if** assignments unchanged from previous iteration **then**
15:             **break**                        $\triangleright$ Converged
16:         **end if**
17:         **for** each cluster $k = 1$ to $c$ **do**
18:             Collect assigned vectors: $\mathcal{B}_k = \{\mathbf{b}_i \mid z_i = k\}$
19:             **if** $|\mathcal{B}_k| > 0$ **then**
20:                 Compute dimension-wise mean: $\mathbf{m}_k = \frac{1}{|\mathcal{B}_k|} \sum_{\mathbf{b}_i \in \mathcal{B}_k} \mathbf{b}_i$
21:                 Update centroid by majority vote: $\mathbf{c}_k = \text{sign}(\mathbf{m}_k)$
22:                 Resolve ties (zeros) by setting to $+1$: $\mathbf{c}_k[\mathbf{m}_k = 0] \leftarrow 1$
23:             **end if**
24:         **end for**
25:     **end for**
26:     Reconstruct binary matrix: $\hat{\mathbf{B}} = \text{VECTOR\_TO\_WEIGHT}(\mathbf{C}[z], \mathbf{B})$
27:     Compute loss: $\mathcal{L} = \|\mathbf{W} - (\alpha \cdot \hat{\mathbf{B}} + \mu) \cdot \text{mask}\|_2^2$
28:     **return** $\{\mathbf{C}, z, \hat{\mathbf{B}}, \mathcal{L}\}$
29: **end function**

---

**Algorithm 4** Binary Vector Processing

---

1: **function** WEIGHT_TO_VECTOR($\mathbf{B}$, $v$)
2:      Extract non-zero elements from $\mathbf{B}$
3:      Pad with alternating $+1/-1$ to ensure divisibility by $v$
4:      Reshape to form vectors of length $v$
5:      **return** Vectors of shape $[N, v]$
6: **end function**
7: **function** VECTOR_TO_WEIGHT($\mathbf{V}$, $\mathbf{B}$)
8:      Create mask of non-zero positions in $\mathbf{B}$
9:      Flatten vectors and remove padding
10:     Place vector elements back into original positions
11:     **return** Reconstructed binary matrix
12: **end function**

---

## C.2    BINARY CODEBOOK OPTIMIZATION

The core of our approach is an EM-based algorithm optimized specifically for binary weights refer to pseudocode 3. Hamming distance can be calculated by $d_H(\mathbf{b}, \mathbf{c}) = \text{POPCNT}(\mathbf{b} \oplus \mathbf{c})$, and sign base centroid update can be accelerated by POPCNT, PCMPGTB.

## C.3    EFFICIENT IMPLEMENTATION DETAILS

Our implementation incorporates several optimizations specifically for binary weights:

---

**Algorithm 5** Efficient Binary Vector Packing and Unpacking

1: **function** WEIGHT_TO_VECTOR($\mathbf{B}$, $v$)
2:     Extract indices of non-zero entries: idx $\leftarrow \{(i, j) \mid \mathbf{B}_{i,j} \neq 0\}$
3:     Extract binary values and map: $b \leftarrow (\mathbf{B}_{\text{idx}} + 1)/2 \in \{0, 1\}$
4:     Pad $b$ with 0/1 alternately to make length divisible by $v$
5:     Reshape $b$ to bit-vectors: $\mathbf{V}_{\text{bit}} \in \{0, 1\}^{N \times v}$
6:     **return** ($\mathbf{V}_{\text{bit}}$, idx)
7: **end function**
8: **function** VECTOR_TO_WEIGHT($\mathbf{V}_{\text{bit}}$, idx)
9:     Flatten bits: $b \leftarrow \text{reshape}(\mathbf{V}_{\text{bit}}, [-1])$
10:     Remove padding to match len(idx)
11:     Map bits back: $\mathbf{B}_{\text{idx}} \leftarrow 2 \cdot b - 1$
12:     Fill remaining entries in $\mathbf{B}$ with zeros
13:     **return** $\mathbf{B}$
14: **end function**

---

Table 6: Complete comparison of the perplexity score on WikiText2 and averaged accuracy on Zero-shot Common Sense Reasoning tasks for LLaMA Model Family

| Models | Method | #Bits W-A-KV | Winogrande | OBQA | Hellaswag | Boolq | ARC-e | ARC-c | RTE | Average | WikiText2 |
|---|---|---|---|---|---|---|---|---|---|---|---|
| | FP16 | 16-16-16 | 69.93 | 43.80 | 76.20 | 74.98 | 72.90 | 44.71 | 67.15 | 64.37 | 5.68 |
| | BTC-LLM | 1.11-16-16 | 68.98 | 40.6 | 71.49 | 73.79 | 68.6 | 40.87 | 63.9 | 61.18 | 6.23 |
| LLaMA-1-7B | BTC-LLM | 0.90-16-16 | 68.9 | 74.4 | 71.44 | 74.4 | 69.65 | 40.53 | 60.29 | 60.86 | 6.24 |
| | BTC-LLM | 0.80-16-16 | 67.4 | 69.02 | 68.79 | 69.02 | 64.6 | 37.97 | 50.18 | 56.71 | 6.72 |
| | BTC-LLM | 0.70-16-16 | 56.99 | 31.2 | 49.64 | 63.49 | 46.46 | 27.22 | 53.43 | 46.92 | 10.72 |
| | FP16 | 16-16-16 | 72.69 | 33.20 | 59.91 | 77.89 | 77.40 | 46.42 | 70.40 | 63.80 | 5.09 |
| | BTC-LLM | 1.11-16-16 | 72.77 | 43.2 | 75.57 | 75.5 | 72.94 | 44.8 | 67.87 | 64.66 | 5.53 |
| LLaMA-1-13B | BTC-LLM | 0.90-16-16 | 71.43 | 44.4 | 75.12 | 77.34 | 72.94 | 43.94 | 69.31 | 64.93 | 5.56 |
| | BTC-LLM | 0.80-16-16 | 67.4 | 69.02 | 68.79 | 69.02 | 64.6 | 37.97 | 50.18 | 60.82 | 6.01 |
| | BTC-LLM | 0.70-16-16 | 63.54 | 33.6 | 54.75 | 66.51 | 54.17 | 30.8 | 52.71 | 50.87 | 9.01 |
| | FP16 | 16-16-16 | 75.69 | 48.8 | 82.59 | 82.66 | 78.83 | 52.73 | 67.15 | 69.78 | 4.10 |
| | BTC-LLM | 1.11-16-16 | 74.74 | 47.6 | 79.94 | 81.83 | 78.07 | 50.94 | 66.79 | 68.56 | 4.59 |
| LLaMA-1-30B | BTC-LLM | 0.90-16-16 | 74.82 | 46.8 | 79.82 | 78.13 | 77.78 | 51.19 | 63.18 | 67.39 | 4.63 |
| | BTC-LLM | 0.80-16-16 | 73.16 | 45.0 | 76.07 | 71.71 | 73.99 | 45.39 | 66.06 | 64.48 | 5.29 |
| | BTC-LLM | 0.70-16-16 | 67.8 | 36.0 | 58.93 | 65.87 | 62.84 | 37.12 | 54.51 | 54.72 | 7.80 |
| | FP16 | 16-16-16 | 77.11 | 47.2 | 84.15 | 84.86 | 79.84 | 55.55 | 69.68 | 71.2 | 3.53 |
| | BTC-LLM | 1.11-16-16 | 76.56 | 45.8 | 82.09 | 84.37 | 79.25 | 53.84 | 69.31 | 70.17 | 3.94 |
| LLaMA-1-65B | BTC-LLM | 0.90-16-16 | 76.01 | 46.6 | 81.79 | 82.94 | 79.17 | 54.1 | 71.84 | 70.35 | 4.03 |
| | BTC-LLM | 0.80-16-16 | 74.98 | 45.4 | 78.77 | 76.76 | 77.31 | 50.94 | 66.79 | 67.28 | 4.74 |
| | BTC-LLM | 0.70-16-16 | 70.01 | 40.4 | 65.66 | 71.04 | 66.75 | 40.02 | 60.65 | 59.22 | 6.61 |
| | FP16 | 16-16-16 | 68.67 | 44.2 | 75.93 | 77.86 | 74.62 | 46.25 | 63.54 | 64.44 | 5.47 |
| | BTC-LLM | 1.11-16-16 | 67.09 | 41.4 | 71.36 | 74.71 | 71.17 | 41.47 | 65.7 | 61.84 | 6.06 |
| | BTC-LLM | 0.90-16-16 | 67.64 | 41.0 | 71.35 | 74.16 | 68.9 | 39.51 | 63.18 | 60.82 | 6.07 |
| | BTC-LLM | 0.80-16-16 | 74.98 | 45.4 | 78.77 | 76.76 | 77.31 | 50.94 | 66.79 | 67.28 | 6.60 |
| LLaMA-2-7B | BTC-LLM | 0.80-8-16 | 65.75 | 39.2 | 67.94 | 73.09 | 69.82 | 39.33 | 62.09 | 59.6 | 6.61 |
| | BTC-LLM | 0.80-8-8 | 65.98 | 27.00 | 50.06 | 71.77 | 71.13 | 36.38 | 62.09 | 59.8 | 6.52 |
| | BTC-LLM | 0.80-4-16 | 63.3 | 38.0 | 65.42 | 68.53 | 61.32 | 36.6 | 57.04 | 55.74 | 7.20 |
| | BTC-LLM | 0.80-4-4 | 58.17 | 22.40 | 44.90 | 67.58 | 63.05 | 30.55 | 57.04 | 53.44 | 7.94 |
| | BTC-LLM | 0.70-16-16 | 58.88 | 33.6 | 48.84 | 62.45 | 47.14 | 28.07 | 51.26 | 47.18 | 11.02 |
| | FP16 | 16-16-16 | 72.22 | 45.4 | 79.39 | 80.58 | 77.48 | 49.32 | 64.98 | 67.05 | 4.88 |
| | BTC-LLM | 1.11-16-16 | 71.11 | 44.8 | 75.24 | 76.79 | 74.66 | 45.31 | 62.82 | 64.39 | 5.29 |
| LLaMA-2-13B | BTC-LLM | 0.90-16-16 | 71.9 | 45.0 | 75.4 | 76.21 | 74.79 | 46.33 | 62.45 | 64.58 | 5.32 |
| | BTC-LLM | 0.80-16-16 | 69.46 | 41.6 | 72.63 | 71.53 | 70.75 | 42.75 | 64.62 | 61.91 | 5.83 |
| | BTC-LLM | 0.70-16-16 | 62.83 | 32.8 | 52.07 | 63.18 | 54.12 | 30.89 | 51.99 | 49.7 | 8.76 |
| | FP16 | 16-16-16 | 73.01 | 44.6 | 79.06 | 81.16 | 77.82 | 53.41 | 68.23 | 68.18 | 6.13 |
| | BTC-LLM | 1.11-16-16 | 72.77 | 42.8 | 73.53 | 76.94 | 73.02 | 47.01 | 59.93 | 63.71 | 7.70 |
| LLaMA-3-8B | BTC-LLM | 0.90-16-16 | 72.69 | 43.0 | 73.53 | 77.4 | 73.27 | 45.82 | 58.12 | 63.4 | 7.84 |
| | BTC-LLM | 0.80-16-16 | 67.96 | 41.6 | 66.76 | 75.32 | 65.32 | 41.13 | 57.04 | 59.3 | 9.49 |
| | BTC-LLM | 0.70-16-16 | 55.17 | 29.4 | 43.47 | 61.8 | 43.43 | 26.19 | 53.07 | 44.65 | 18.54 |

1. **Early termination**: For cases where the number of unique vectors is less than or equal to the codebook size, we achieve perfect reconstruction with exact vector matching in a single iteration.

Table 7: Complete comparison of the perplexity score on WikiText2 and averaged accuracy on Zero-shot Common Sense Reasoning tasks for Qwen Model Family

| Models | Method | #Bits W-A-KV | Winogrande | OBQA | Hellaswag | Boolq | ARC-e | ARC-c | RTE | Average | WikiText2 |
|--------|--------|------|-----------|------|-----------|-------|-------|-------|-----|---------|-----------|
| | FP16 | 16-16-16 | 68.59 | 42.4 | 73.55 | 76.88 | 73.27 | 46.93 | 75.09 | 65.24 | 8.03 |
| | BTC-LLM | 1.11-16-16 | 66.69 | 39.4 | 66.32 | 75.14 | 70.75 | 42.75 | 78.34 | 62.77 | 9.70 |
| Qwen-2.5-3B | BTC-LLM | 0.90-16-16 | 67.96 | 39.4 | 65.9 | 73.27 | 66.29 | 41.89 | 63.9 | 59.8 | 9.85 |
| | BTC-LLM | 0.80-16-16 | 64.88 | 37.0 | 61.54 | 64.92 | 67.21 | 39.68 | 55.96 | 55.88 | 11.26 |
| | BTC-LLM | 0.70-16-16 | 56.27 | 34.0 | 46.98 | 60.12 | 46.68 | 28.58 | 52.71 | 46.48 | 18.71 |
| | FP16 | 16-16-16 | 75.22 | 45.0 | 82.96 | 85.23 | 79.21 | 58.7 | 79.42 | 72.25 | 5.29 |
| | BTC-LLM | 1.11-16-16 | 76.01 | 46.0 | 79.37 | 86.3 | 82.83 | 57.76 | 81.23 | 72.79 | 6.49 |
| Qwen-2.5-14B | BTC-LLM | 0.90-16-16 | 75.53 | 43.8 | 79.12 | 87.28 | 80.47 | 55.97 | 78.34 | 71.5 | 6.58 |
| | BTC-LLM | 0.80-16-16 | 74.43 | 41.2 | 75.42 | 86.02 | 76.64 | 50.0 | 70.4 | 67.73 | 7.42 |
| | BTC-LLM | 0.70-16-16 | 62.98 | 35.0 | 60.11 | 69.05 | 68.56 | 37.12 | 66.06 | 56.98 | 12.28 |
| Qwen-3-0.6B | FP16 | 16-16-16 | 56.43 | 31.4 | 47.3 | 63.82 | 55.93 | 33.7 | 53.79 | 48.91 | 20.95 |
| | BTC-LLM | 0.8-16-16 | 50.2 | 26.6 | 32.61 | 61.16 | 33.42 | 24.66 | 53.07 | 40.25 | 120.08 |
| Qwen-3-1.7B | FP16 | 16-16-16 | 61.17 | 36.6 | 60.46 | 77.68 | 69.95 | 42.75 | 70.04 | 59.81 | 16.71 |
| | BTC-LLM | 1.11-16-16 | 55.41 | 30.6 | 46.02 | 62.17 | 45.96 | 27.3 | 53.43 | 45.84 | 32.56 |
| | FP16 | 16-16-16 | 67.72 | 41.8 | 75.02 | 86.64 | 80.93 | 56.23 | 77.98 | 69.47 | 9.72 |
| Qwen-3-8B | BTC-LLM | 1.11-16-16 | 65.67 | 39.6 | 67.02 | 81.38 | 76.68 | 50.17 | 77.62 | 65.45 | 11.60 |
| | BTC-LLM | 0.90-16-16 | 67.8 | 38.2 | 66.29 | 84.01 | 75.63 | 49.15 | 77.62 | 65.53 | 11.70 |
| | FP16 | 16-16-16 | 73.16 | 46.4 | 78.97 | 89.45 | 82.91 | 60.49 | 77.62 | 72.71 | 8.64 |
| Qwen-3-14B | BTC-LLM | 1.11-16-16 | 67.64 | 40.0 | 66.92 | 85.72 | 71.72 | 48.38 | 78.34 | 65.53 | 12.05 |
| | BTC-LLM | 0.90-16-16 | 66.38 | 38.0 | 65.82 | 83.82 | 67.85 | 43.77 | 72.92 | 62.65 | 12.93 |

2. **Efficient centroid updates**: Unlike traditional k-means requiring reconstruction for each update, our method directly computes means and applies the sign function to maintain binary constraints.

3. **Vectorized operations**: We leverage PyTorch's efficient tensor operations like `scatter_add_` and `bincount` to accelerate cluster assignment and centroid updates.

4. **Binary-specific distance metric**: Distance calculations between binary vectors utilize squared Euclidean distance, which is more efficient than computing full reconstruction error.

## C.4 COMPLETE BINARY TRANSFORMATION AND COMPRESSION

Our complete binary transformation and compression (BTC) approach combines learned transformations with binary codebook compression refer to pseudocode 6.

---

**Algorithm 6** Binary Transformation and Compression

---

1: **function** BTC($\mathbf{W}$, [$\mathbf{R}$, $\mathbf{s}$, $\mathbf{d}$])
2:     Apply transformation: $\mathbf{W} \leftarrow \text{diag}(\mathbf{s} \odot \mathbf{d})^{-1} \cdot \mathbf{R}^\top \cdot \mathbf{W}$
3:     Binarize weights: $\alpha, \mathbf{B}, \mu \leftarrow \text{ARB}(\mathbf{W})$
4:     Generate codebook: $\text{idx}, \mathbf{C} \leftarrow \text{BINARYCODEBOOK}(\mathbf{B})$
5:     Reconstruct binary: $\hat{\mathbf{B}} \leftarrow \mathbf{C}[\text{idx}]$
6:     Dequantize: $\hat{\mathbf{W}} \leftarrow \alpha \cdot \hat{\mathbf{B}} + \mu$
7:     **return** $\hat{\mathbf{W}}$
8: **end function**

---

This approach achieves a compression ratio of approximately $16 \cdot v / \lceil \log_2 c \rceil$, providing significant memory savings while maintaining model quality through tailored binary-specific optimization methods.

## D FUTURE WORK ON ACTIVATION AND KV CACHE QUANTIZATION

Activation quantization reduces memory transfer overhead and leverages efficient low-precision compute units. Moreover, we observe substantial redundancy in the KV cache, enabling aggressive low-bit quantization. We further implement KV cache quantization to exploit this potential. First, we redesign the saliency metric for the binary quantizer. Since the KV cache exhibits a shift in

window importance, we assign higher salient weights to local windows. To avoid dequantization overhead from extreme codebook compression, we preserve local windows binary representation without sub-bit quantization.

Given the need for on-the-fly quantization and dequantization in KV cache compression, developing simpler and more computationally efficient quantizers remains an important direction for future research. Inspired by Binarized Neural Networks (BNNs) in convolutional architectures, where activations are also quantized to binary, we aim to further explore fully binarized LLMs with binary activations.

## E  BINARY CODEBOOK ANALYSIS

Finding the optimal binary codebook is **NP-hard**, as it reduces to a special case of the well-known **k-means clustering** problem, which is NP-hard when the number of clusters $K \geq 2$ and vector dimension $D \geq 2$.

In our setting, each codebook vector is constrained to binary values $\{-1, +1\}^D$, and the goal is to choose $K$ such vectors to minimize the total reconstruction error. This requires searching over all possible combinations of $K$ vectors from a space of $2^D$ candidates, yielding:

$$\text{Search space size} = \binom{2^D}{K}, \quad \text{and total complexity: } O\left(\binom{2^D}{K} \cdot N \cdot K \cdot D\right),$$

where $N$ is the number of weight vectors being quantized. This combinatorial explosion makes the global optimum intractable even for moderate $D$, a hallmark of NP-hard problems.

## F  FULL RESULTS

### F.1  QUANTITATIVE RESULTS

In this section, we provide a comprehensive presentation of our results across various datasets to complement the main paper. Specifically, the results include: Complete comparison of the perplexity score on WikiText2 and averaged accuracy on zero-shot common sense reasoning tasks on LLaMA Model Family in Table 6 and Qwen Model Family in Table 7. And validate the effectiveness the activation quantization and KV cache quantization of BTC-LLM.

### F.2  VISUALIZATION RESULTS

Figure 6 and Figure 7 illustrate the relative quantization error between quantized and full-precision weights for BTC-LLM, ARB-LLM, and BiLLM, highlighting the improved accuracy of BTC-LLM. In contrast, Figure 8 and Figure 9 visualize the activation distributions across different layers of LLaMA-2-7B before and after applying BTC-LLM, showing how our method suppresses outliers and promotes a more compact activation range.

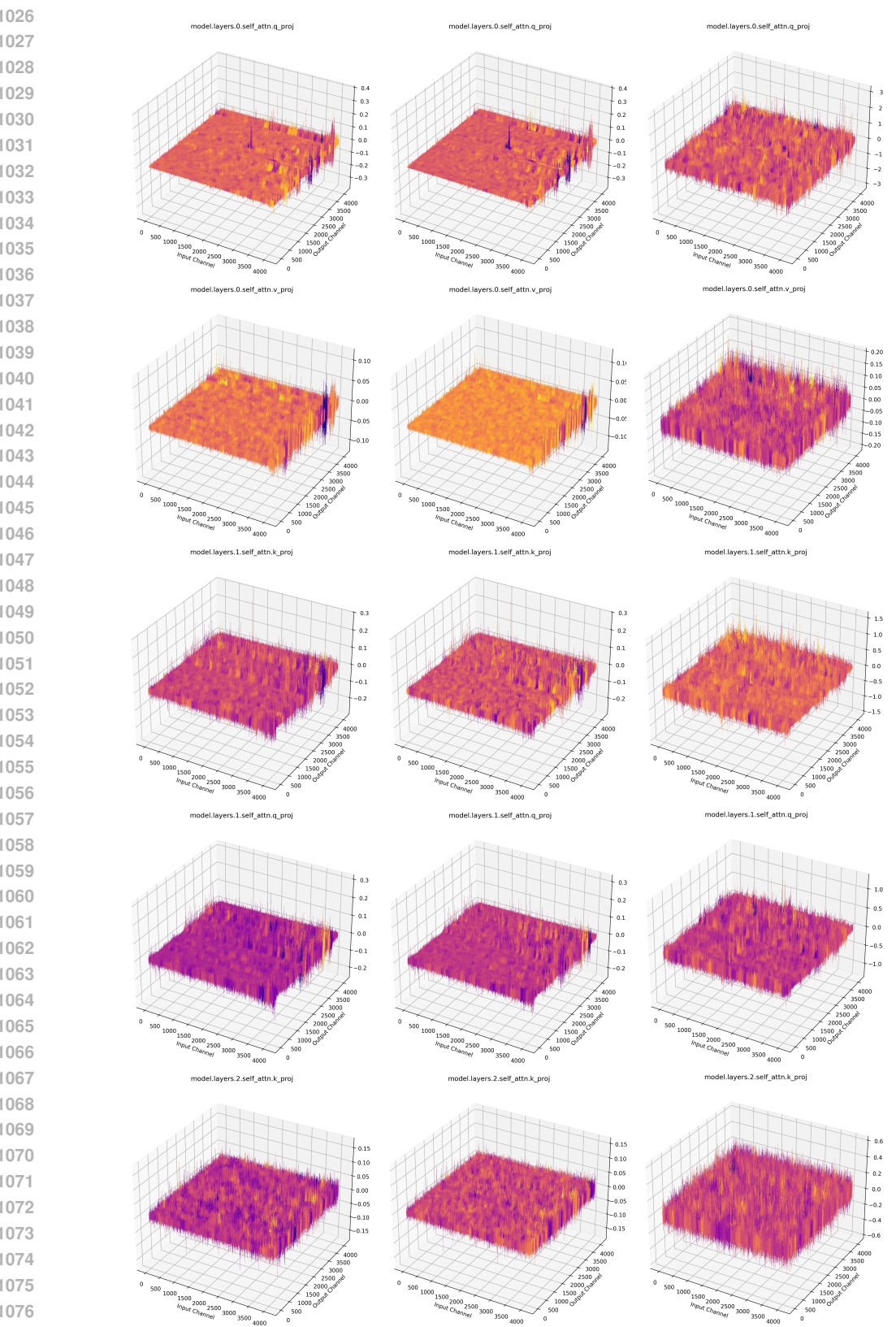

Figure 6: Visualizations comparing of the weight relative quantize error of LLaMA-2-7B with BTC-LLM (1st column), ARB-LLM(2nd column), and BiLLM (3rd column), respectively.

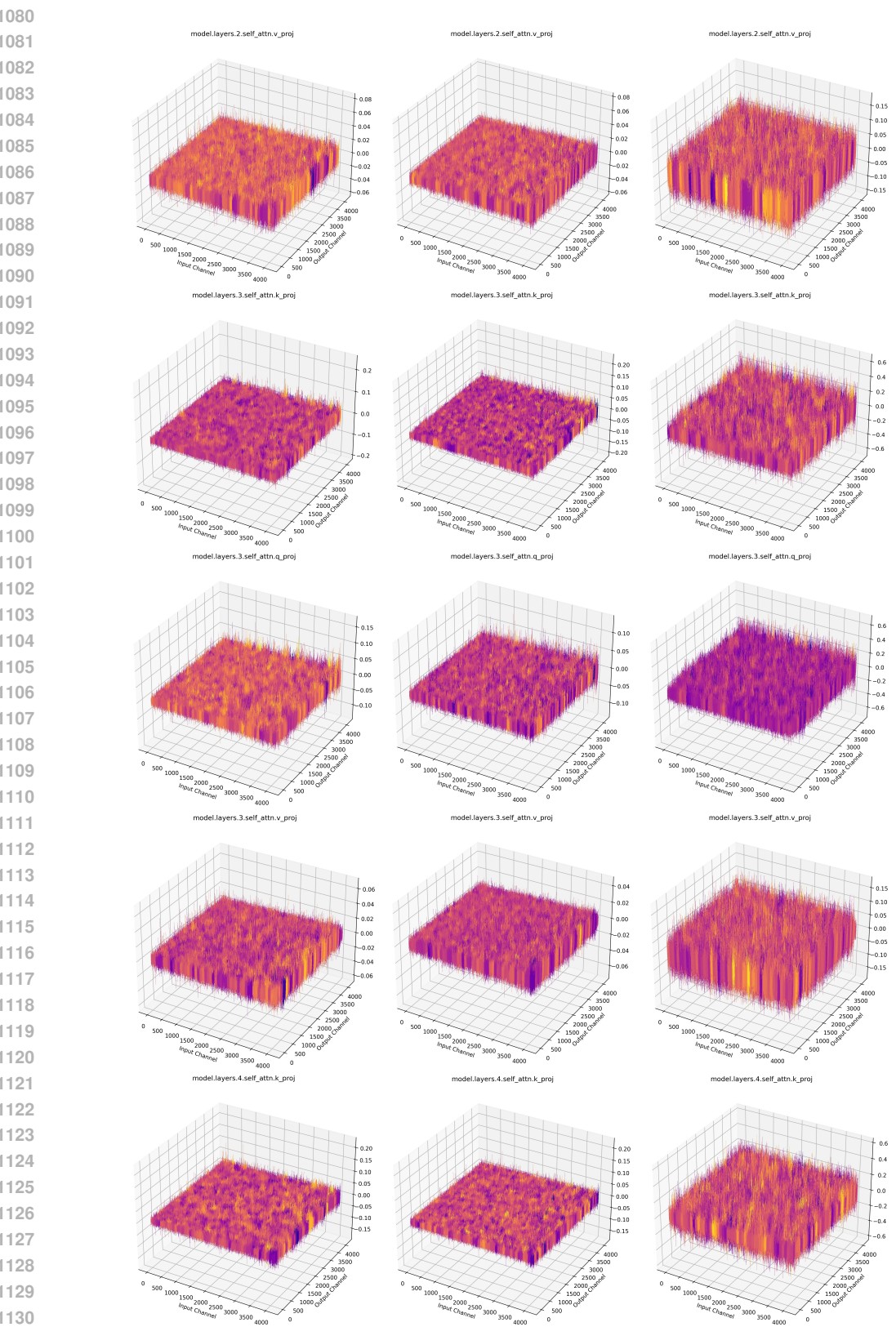

Figure 7: Visualizations comparing of the weight relative quantize error of LLaMA-2-7B with BTC-LLM (1st column), ARB-LLM(2nd column), and BiLLM (3rd column), respectively.

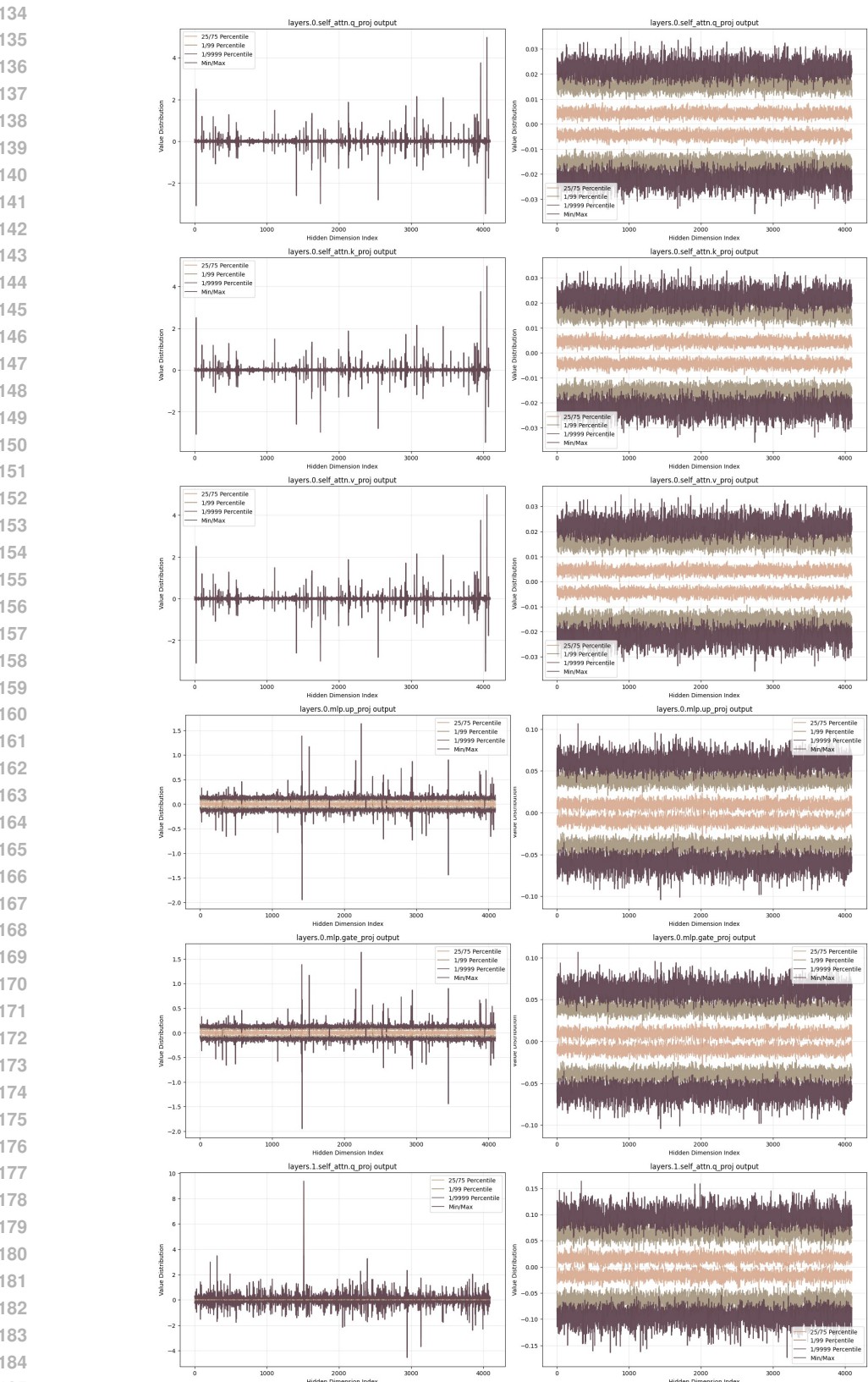

Figure 8: Visualizations of the activation distribution of different layers in LLaMA-2-7B before and after BTC-LLM. Left original activation, Right BTC-LLM activation.

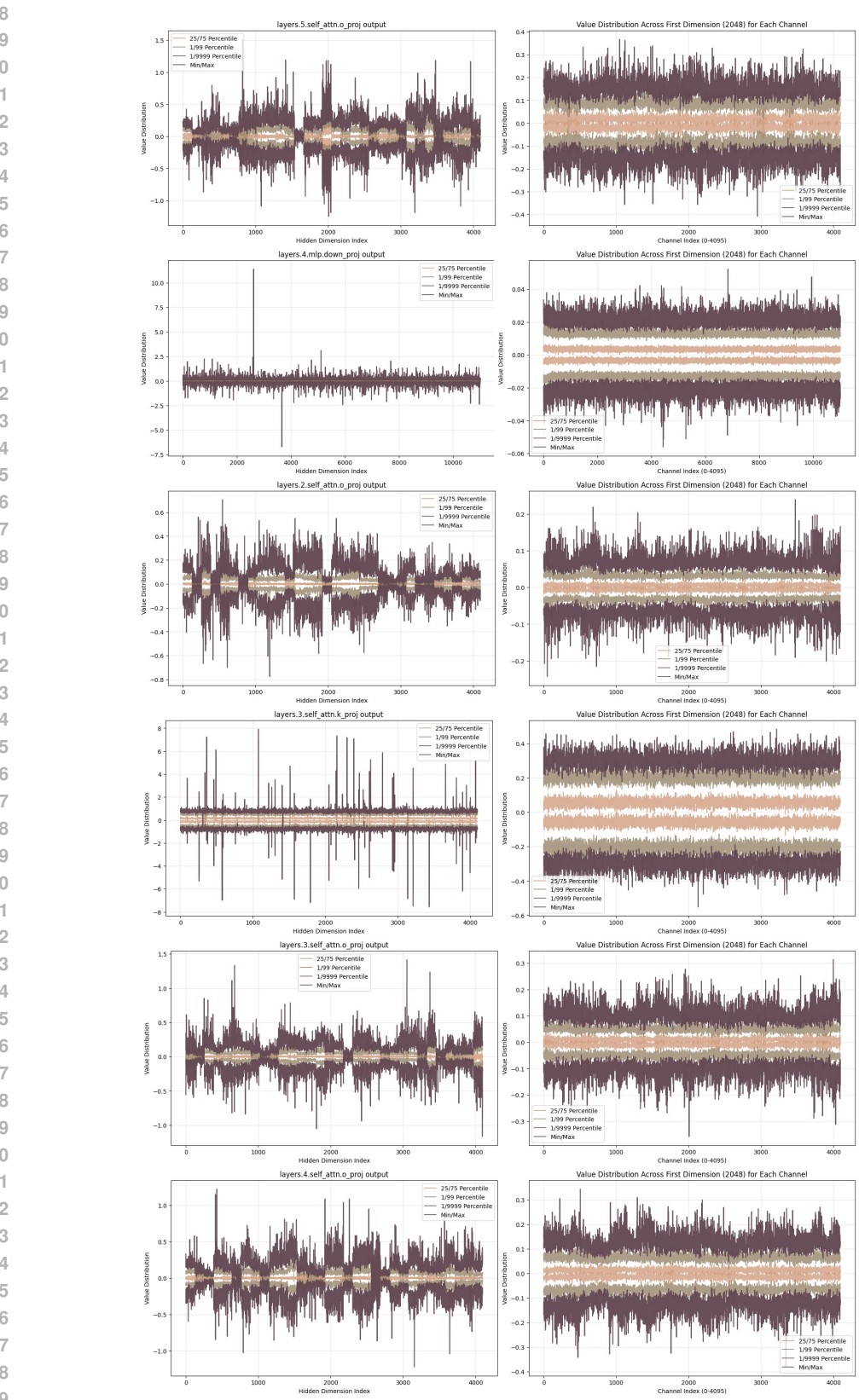

Figure 9: Visualizations of the activation distribution of different layers in LLaMA-2-7B before and after BTC-LLM. Left original activation, Right BTC-LLM activation.

