# OpenReview forum: "BTC-LLM: Efficient Sub-1-Bit LLM Quantization via Learnable Transformation and Binary Codebook"
_ICLR.cc/2026/Conference — ICLR 2026 Conference Withdrawn Submission_

### Official Review · Reviewer_nkwc · 2025-10-24

**Soundness:** 1
**Presentation:** 3
**Contribution:** 1
**Rating:** 0
**Confidence:** 4

**Summary:**

The paper tries to do sub-1-bit quantization.
However, it includes orthogonal proprocessing, which is not accounted for in the memory costs, and after accounting for storing orthogonal matrices, we would have the same storage costs as the original model.

**Strengths:**

Perplexity looks good.

**Weaknesses:**

The paper uses a matrix $R$ of size $n \times m$ for preprocessing before binarization (note that previous works used Hadamard matrices, which are much smaller).

I assume that each weight matrix $W$ has its own orthogonal preprocessing $R$ matrix. For simplicity, I will ignore diagonal factors.
After calculating $R$, we compute $R^TW$ and binarize that and get $B = ARB(R^TW)$ (Algorithm 1).
But for computation of the layer output we need to calculate $y = xRB$, thus we need to store $R$ somehow, but $R$ is as big as the original matrix and thus we would not have any reduction of storage costs.

**Questions:**

How is matrix R stored, and how much space does it take?

---

> ### Author Response · Authors · 2025-11-13
> **On the storage cost of the orthogonal transformation (R)**
>
> **There is no situation orthogonal transform (R) be store during inference in BTC-LLM.**
>
> For simplicity, consider the FFN block. As shown in the pseudo-code below( `BTC_LlamaMLP` and `BinaryLinear`), we use a **shared** orthogonal transform `ffn_trans` for the up, gate, and down projections in each FFN block:
>
> ```python
> self.ffn_trans = Transformation(self.hidden_size, self.intermediate_size)
> self.up_proj   = BinaryLinear(self.up_proj,   self.ffn_trans)
> self.gate_proj = BinaryLinear(self.gate_proj, self.ffn_trans)
> self.down_proj = BinaryLinear(self.down_proj, self.ffn_trans.transpose())
> ```
>
> During training (`fake_quant=True`), we do **not** merge the transform into the weights; `BinaryLinear.forward` applies the transform on the fly and optimizes the quantizer with STE:
>
> ```python
> trans_weight    = self.trans(self.linear.weight)                # W → T W
> binary_weight   = self.Binary_Quantizer(trans_weight, fake_quant=True)
> compress_weight = self.Codebook_compressor(binary_weight, fake_quant=True)
> output          = F.linear(x, compress_weight)
> ```
>
> Before deployment, we call `reparameterize()` once per layer:
>
> ```python
> def reparameterize(self):
>     weight_trans  = self.trans(self.linear.weight)
>     binary_weight = self.Binary_Quantizer(weight_trans, fake_quant=False)
>     codebook, index = self.Binary_codebook_compressor(binary_weight, fake_quant=False)
>     self.register_buffer("codebook", codebook)
>     self.register_buffer("index", index)
>     del self.linear, self.trans
> ```
>
> This folds the transform (T) into the binary weights and stores only their **codebook and indices**. The original full-precision weight `linear.weight` and the transform `trans` are deleted and are **not** present in the deployed model. Inference only uses the LUT kernel over `(codebook, index)`:
>
> ```python
> return self.LUT_kernel(x, self.codebook, self.index)
> ```
>
> Thus orthogonal preprocessing is used solely to construct a better binary representation **offline** and does **not** add any parameters or storage at inference time.
>
>
>
> ---
>
>
>
> ```python
> class BinaryLinear(nn.Module):
>     def __init__():
>         self.linear = linear
>         self.trans = trans
>         self.Binary_Quantizer = ARBQuantizer(args)
>         self.Codebook_compressor = CodebookCompressor(args)
>
>     def forward(self, x, fake_quant=False):
>         if fake_quant:
>             trans_weight = self.trans(self.linear.weight)
>             binary_weight = self.Binary_Quantizer(trans_weight, fake_quant=True) # STE
>             compress_weight = self.Codebook_compressor(binary_weight, fake_quant=True) # STE
>             output = F.linear(x, compress_weight)
>             return output
>         else:
>             # Triton Binary LUT kernel
>             return self.LUT_kernel(x, self.codebook, self.index)
>
>     def reparameterize(self):
>         weight_trans = self.trans(self.linear.weight)
>         binary_weight = self.Binary_Quantizer(weight_trans, fake_quant=False)
>         codebook, index = self.Binary_codebook_compressor(binary_weight, fake_quant=False)
>         self.register_buffer("codebook", codebook)
>         self.register_buffer("index", index)
>         del self.linear, self.trans
>
> class BTC_LlamaMLP(LlamaMLP):
>     def __init__(self, args):
>         # activation quantizer for w1a8, w1a4 setting
>         self.activation_quantizer_gate = ActivationQuantizer(bits=args.a_bits)
>         self.activation_quantizer_up = ActivationQuantizer(bits=args.a_bits)
>         self.activation_quantizer_down = ActivationQuantizer(bits=args.a_bits)
>
>         # shared orthogonal transformation for FFN
>         self.ffn_trans = Transformation(self.hidden_size, self.intermediate_size)
>         self.up_proj = BinaryLinear(self.up_proj, self.ffn_trans, args)
>         self.gate_proj = BinaryLinear(self.gate_proj, self.ffn_trans, args)
>         self.down_proj = BinaryLinear(self.down_proj, self.ffn_trans.transpose() , args)
>         self.fake_quant = False
>
>     def forward(self, x):
>         # reparameterize the whole model before real quant deploy
>         x_up = self.up_proj(self.activation_quantizer_up(x), fake_quant=self.fake_quant)
>         x_gate = self.gate_proj(self.activation_quantizer_gate(x), fake_quant=self.fake_quant)
>         ac = self.act_fn(x_gate)
>         x = x_up * ac
>         x = self.down_proj(self.activation_quantizer_down(x), fake_quant=self.fake_quant)
>         return x
>
>     def set_fake_quant(self, fake_quant=False):
>         self.fake_quant = fake_quant
> ```

---

> ### Comment · Reviewer_nkwc · 2025-11-13
> **Math does not hold up**
>
> 1) On line 303 (and also line 263) you explicitly say, that:
> $x'W' = (xT^T)(TW)$
> Thus, the input must be processed by the inverse of that orthogonal transform (transpose).
>
> 2) In Figure 4, you wrap attention and FFN blocks in those orthogonal transforms. But that would only work if the whole block were linear, which it is not. There is a reason why Quip#, QTIP do not do this.

---

> > ### Author Response · Authors · 2025-11-14
> >
> > Thank you for the detailed follow-up. We now better understand the source of your concerns.
> >
> > **1. On Eq. $x'W' = (xT^{\top})(TW)$ and the loss definition.**
> > You are right that, in the original draft, our notation was misleading. We wrote a linear layer wise loss and also introduced $x'W' = (xT^{\top})(TW)$. This combination indeed looks as if the input must always be processed by $\(T^{\top}\)$, which was not what we intended to convey.
> >
> > In the actual implementation, the optimization is **block-wise**, not per linear layer. In the revised version (lines 261–269, highlighted in blue), we now state this explicitly:
> >
> > $
> > \min_{\mathbf{T}} \Big(  \mathbf{MSE}\big( \mathcal{F}(X) - \hat{\mathcal{F}}(X;\mathbf{T}) \big) + \mathcal{L}_{\text{aux}} \Big),
> > $
> >
> > Actually the linear output $y=xW'=xTW$ is changed, involving $xT^{\top}$ was meant to show that when outputs of a transformer block are unchanged. In the revision we have removed this shorthand.
> >
> > ---
> >
> > **2. On Fig. 4 and wrapping attention, FFN blocks with transforms.**
> > We fully agree that it would be incorrect to apply a single transform to the **entire** nonlinear block. Our method does **not** do that. In the revised text (lines 287–295, blue), we clarify the exact usage:
> >
> > In the attention block, we use a shared transform $T_{\text{attn}}$ for Q/K/V and its inverse for the output projection:
> >
> > $Q' = X W_q T_\text{attn}$, $K' = X W_k T_\text{attn}$, $V' = X W_v T_\text{attn}$, $W_{o} = T_\text{attn}^{-1} W_o.$
> >
> > Because $T_{\text{attn}}$ is orthogonal,
> >
> > **$\mathrm{softmax}(Q'K'^{\top})V'W'_o = \mathrm{softmax}(Q K^{\top}) V W_o$, so the attention output is unchanged.**
> >
> > An analogous paired transform $T_{\text{ffn}}$ is used only for the up/gate/down projections in the FFN block. Thus each block remains functionally equivalent while its weights are reparameterized.
> >
> > After calibration, we **fold the transforms** into the binary weights and discard $T_{\text{attn}}$ and $T_{\text{ffn}}$; only the compressed binary weights are stored and used at inference.
> >
> > We acknowledge that the original notation and figure could easily be read in the way you described, and we have now corrected both the equations and the accompanying explanation to faithfully reflect our actual implementation.
> >
> > We would sincerely appreciate it if you could reconsider your assessment of the soundness and contribution of our work in light of these clarifications and revisions.

---

> > > ### Comment · Reviewer_nkwc · 2025-11-14
> > >
> > > Attention transformation does not hold because you have multiple heads (it would work with a single head).
> > >
> > > FFN transformation does not work. Here is a simple demo:
> > >
> > > ```
> > > import torch
> > > import torch.nn as nn
> > >
> > > ortho = torch.randn(128, 128)
> > > torch.nn.init.orthogonal_(ortho)
> > >
> > > #check
> > > print("check ortho", ortho.matmul(ortho.T))
> > >
> > > class MLP(nn.Module):
> > >     def __init__(self):
> > >         super().__init__()
> > >         self.up = nn.Linear(128, 128, bias=False)
> > >         self.gate = nn.Linear(128, 128, bias=False)
> > >         self.down = nn.Linear(128, 128, bias=False)
> > >
> > >     def forward(self, x):
> > >         # Works with just linear layers stacked
> > >         # return self.down(self.up(x))
> > >         # Does not work even without relu
> > >         # return self.down(self.up(x) * (self.gate(x)))
> > >         return self.down(self.up(x) * torch.relu(self.gate(x)))
> > >
> > > mlp = MLP()
> > >
> > > print("mlp expected", mlp(test_inp))
> > >
> > > # Add orthogonalization
> > > mlp.up.weight.data = ortho.matmul(mlp.up.weight)
> > > mlp.gate.weight.data = ortho.matmul(mlp.gate.weight)
> > > mlp.down.weight.data = mlp.down.weight.matmul(ortho.T)
> > >
> > > print("mlp after ort", mlp(test_inp))
> > > ```
> > >
> > > There is a nonlinear transformation in the middle; there is no way that an orthogonal transformation can be folded here.

---

### Official Review · Reviewer_dCEk · 2025-10-31

**Soundness:** 1
**Presentation:** 1
**Contribution:** 1
**Rating:** 2
**Confidence:** 3

**Summary:**

The paper introduces a new method for sub-1-bit quantization, built on top of 1-bit quantization of the weights. Matrices of 1-bit quantized weights are approximated using a coodebook of K-centroids over sub-vectors of rows. To reduce outliers and improve quantization, the authors further propose to learn an invertible transformation for the weight matrices, with an auxiliary loss guiding toward an improved representation.

**Strengths:**

- The paper provides extensive details about the codebook assignment method.
- The proposed method improves scores across all tasks compared to STBLLM on LLaMA-1 and LLaMA-2.
- The proposed method achieved low perplexity across all models, even at very low bitrates.

**Weaknesses:**

- The major weakness is the lack of comparisons with existing methods on the more challenging settings. Application on downstream task is only superficially experimented in the main paper, with comparison only with STBLLM and FP16, on old models (LLaMA-1 and LLaMA-2), while table 6 shows clearly that LLaMA-3 is the most challenging compression setting. Both this lack of experimental settings and comparison with baselines severely hinders the results of this work.
- More effort is put into comparing perplexity, but this only acts as a proxy for actual model performance on tasks.
- It is unclear what is the novelty of the outlier elimination part compared to existing methods.
- The paper is hard to follow. Although the overall structure is well organized, each part is unclear, lack details and clarity. Some sentences are not complete, and the paper overall it lacks polishing.
- In the preliminaries, explanations are unclear, especially for the “Codebook Compression” part. This makes it difficult to understand later how the codebook quantization is applied. There is also no reference to existing literature in this section, which would help understand more easily this part.
- The paper is not clear on how the orthogonal transformation is merged into the weight matrix. Figure 4 seems to hint that some transformations are shared by multiple weight matrices, while this is not explicit in the paper, and there is no clear explanation on how the merging of the inverse matrix occurs, especially when combined to a block with shared orthogonal transformations.

**Questions:**

- Why introduce both $R$ and $D_{\pm}$, which are both orthogonal matrices? What does the second term adds to the expressivity?
- How does your model compare on downstream tasks compared to other quantization methods on LLama-3 or Qwen3?
- Could you clarify your contribution and method?

---

### Official Review · Reviewer_mbFC · 2025-10-31

**Soundness:** 3
**Presentation:** 2
**Contribution:** 3
**Rating:** 6
**Confidence:** 3

**Summary:**

This paper proposes a system-oriented method that compresses LLM weights to sub-1-bit while remaining fast and deployable on hardware. It applies a binary codebook-based vector quantization. To reduce distribution mismatch between real-valued weights and the binary codebook space, the method incorporates a learnable transformation. It uses an EM-style training loop alternating E-step index updates and M-step codebook/transform updates.  The empirical evaluation is strong, reporting real memory and latency gains with solid ablations across bitwidths and components.

**Strengths:**

1. The performance gain is consistent across models and tasks.
2. The paper not only shows perplexity gains, and reductions in memory and measured speedups on hardwares.
3. The ablation studies are comprehensive.

**Weaknesses:**

1. Section 4.2 is hard to follow. Is this transformation entirely a new design, or does it draw on prior lines of work?

**Questions:**

1. Could the author compare the quantization and mean accuracy under different setup with other VQ-based baselines?
2. Could the author include AQLM as a baseline?

---

### Official Review · Reviewer_A7dv · 2025-11-04

**Soundness:** 3
**Presentation:** 2
**Contribution:** 3
**Rating:** 6
**Confidence:** 4

**Summary:**

This paper introduces BTC-LLM, a sub-1-bit quantization method for LLMs that combines a binary codebook and a learnable transformation to improve both compression and accuracy. The approach avoids sparse masks and instead clusters binary weight patterns while using a scaling–rotation transform to align binarized weights with full-precision ones. Experiments on Llama and Qwen models show strong results with minimal accuracy loss and efficiency gains.

**Strengths:**

- The combination of a binary codebook and learnable transformation is smart. It tackles both redundancy and activation outliers which are two core issues in binary quantization.
- The paper provides detailed formulations, including the codebook optimization (binary K-Means style) and orthogonal transformation training (Cayley SGD). The authors explain computational trade-offs and hardware friendliness.
- The discussion on implementing the codebook via XNOR-POPCNT operations and shared-memory caching is practical. The demonstrated latency and memory reductions sound good.

**Weaknesses:**

Areas for improvement:

- Some sections (especially the methodology) are quite dense, with long formulae and mixed pseudo-code, which may hinder readability. Visual aids for algorithm flow could help.
- The authors briefly mention the lack of KV-cache compression and the overhead of the learnable transform only in the appendix. A more upfront treatment of these trade-offs would make the contribution sound more balanced.
- While the results are extensive, there’s limited reporting on real downstream application tasks (e.g., instruction following or reasoning benchmarks). Zero-shot QA tasks are useful but don’t fully stress model robustness under binary compression.
- Although speedups are claimed (1.6× over FP16), there is little granular profiling (e.g., GPU utilization, kernel fusion comparisons). More system-level measurements would strengthen the argument for hardware efficiency.

**Questions:**

- How much extra compute or memory does the learnable transformation add during quantization or inference? A short breakdown would help gauge practicality.
- Have you tried BTC-LLM on instruction-following or reasoning tasks like MMLU or GSM8K? It’d show how well it generalizes beyond zero-shot QA.
- You mention sharing a codebook across layers. how much accuracy does that save or cost? Would per-layer codebooks improve results?
- The 1.6× speedup is great, but could you share a bit more detail, e.g., GPU utilization or whether any custom kernels were used?
- How does BTC-LLM stack up against recent 2-bit or ternary methods in terms of both accuracy and energy efficiency?

---

### Note · Authors · 2025-11-18

I have read and agree with the venue's withdrawal policy on behalf of myself and my co-authors.